# The Vault: A Comprehensive Multilingual Dataset for Advancing Code Understanding and Generation

**Dung Nguyen Manh**[1,*]**, Nam Le Hai**[1,3,*]**, Anh T. V. Dau**[1,3]**,**
**Anh Minh Nguyen**[1]**, Khanh Nghiem**[1]**, Jin Guo**[4,5]**, Nghi D. Q. Bui**[2]

[1]FPT Software AI Center
`{dungnm31, namlh35, anhdtv7, minhna4, khanhnv22}@fpt.com`

[2]Fulbright University, Viet Nam
`nghi.bui@fulbright.edu.vn`

[3]Hanoi University of Science and Technology, Viet Nam
[4]School of Computer Science, McGill University, Canada
[5]Mila - Quebec AI Institute

## Abstract

We present The Vault, a dataset of high-quality code-text pairs in multiple programming languages for training large language models to understand and generate code. We present methods for thoroughly extracting samples that use both rule-based and deep learning-based methods to ensure that they contain high-quality pairs of code and text, resulting in a dataset of 43 million high-quality code-text pairs. Our extensive evaluations on common coding tasks including code generation, code search and code summarization show that when fine-tuning Code Large Language Models on The Vault, such models outperform the same models trained on other datasets such as Code-SearchNet. We also provide detailed analyses of our datasets to assess the effects of various programming languages and docstrings on the performance of such models.

## 1  Introduction

The advent of deep learning and advancements in large language models (LLMs) have spurred a revolution in the field of code representation learning. These developments, supported by the growing accessibility of vast open-source code repositories, have heralded the emergence of code large language models (CodeLLMs) for code generation and understanding tasks. The sheer volume of these repositories and the rich, unprocessed raw data they contain, serve as unparalleled resources for training LLMs. Consequently, current state-of-the-art models for coding tasks effectively utilize

these expansive datasets for training. However, it is important to note that these datasets, including The Stack [Kocetkov et al., 2022] and The Pile [Gao et al., 2020a], often comprise unprocessed data.

Alternatively, there are established datasets, such as CONCODE [Iyer et al., 2018b], FunCom [LeClair et al., 2019], Deepcom [Hu et al., 2020] for code summarization tasks; APPS [Hendrycks et al., 2021] for text-to-code generation; and Code-SearchNet [Husain et al., 2019] for code search. These datasets contain carefully curated code-text pairs. Although considerably smaller in comparison to raw code datasets (e.g., 2.3M functions in CodeSearchNet [Husain et al., 2019] versus 197M files in The Stack [Kocetkov et al., 2022]), they provide high-quality code-text pairings that significantly enhance the effectiveness of model training.

Consequently, we identify two main types of datasets used to train CodeLLMs: large yet unprocessed, and smaller yet well-structured (e.g., arranged into code-text pairs). The scaling law [Kaplan et al., 2020, Gordon et al., 2021, Sorscher et al., 2022] indicates that the volume of training data is crucial for model performance. However, other studies underscore the importance of dataset quality over quantity in training superior LLMs [Zhou et al., 2023, Sorscher et al., 2022, Dau et al., 2022, Brown et al., 2020, Khan et al., 2020]. Given these observations, we propose that an ideal dataset for training CodeLLMs should combine both elements: it should be expansive in volume and meticulously processed to ensure quality.

In this paper, we present The Vault dataset, detailing its creation process, the toolkit developed

---

[*]Equal contribution

for constructing and quality-controlling code-text pairs from raw source code, as well as an analysis of The Vault's metrics. We also share empirical results obtained from utilizing The Vault to fine-tune well-known foundational models. Our specific contributions include the following:

- A dataset with approximately 43M pairs of high-quality code-text pairs (over 10 times larger than CoDesc), 243M unimodal samples, and 69M pairs of line comments with context from 10 popular programming languages (Java, JavaScript, Python, Ruby, Rust, Golang, C#, C++, C, PHP), more diverse than CodeSearchNet, which has six programming languages.

- A novel approach to use a pre-trained language model for detecting and removing noisy samples to complement traditional rule-based methods.

- A thorough process for transforming raw source code into code-text pairs and filtering noisy samples. We have released the toolkit used in this process to the open community via a public GitHub repository[1], including tools for parsing code and docstrings in different programming languages.

- We perform extensive evaluation where we fine-tuned different CodeLLMs with The Vault compared to other datasets, such as CodeSearchNet on various code understanding tasks, including code generation, code summarization and code search. The results show that models fine-tuned on The Vault outperform those fine-tuned on CodeSearchNet (code summarization, code search) and outperform the original model by a significant margin (code generation on pass@k over HumanEval and MBPP datasets).

## 2 Related works

**Code Large Language Models for Understanding and Generation**   Code large language models facilitate various code understanding and code generation tasks, including but not limited to code generation [Feng et al., 2020a, Wang et al., 2023, Elnaggar et al., 2021, To et al., Luo et al., 2023, Shen et al., 2023], code completion [Feng et al., 2020a, Wang et al., 2023, Peng et al., 2021], program repair [Xia et al., 2022], program classification [Bui et al., 2021a,c,b] and code translation [Roziere

et al., 2020, Bui et al., 2019]. A significant portion of recent research employs language models, originally developed for natural language processing, for handling code [Feng et al., 2020a, Wang et al., 2023, Guo et al., Ahmad et al., 2021b, Bui et al., 2021b, Elnaggar et al., 2021, Peng et al., 2021, Kanade et al., 2020, Chakraborty et al., 2022, Ahmed and Devanbu, 2022, Niu et al., 2022]. Such approaches largely regard code as analogous to text and adapt pretraining strategies that mirror those used for natural languages. CodeBERT [Feng et al., 2020a], for instance, modifies a Roberta model [Liu et al., 2019] to pretrain a code model on multiple programming languages. CodeT5 [Wang et al., 2021] and CodeT5+ [Wang et al., 2023] employs unique identifier information from source code to pretrain the T5 model [Raffel et al., 2019] for code in a multi-modal fashion.

**Datasets for Code Representation Learning:** Code is commonly represented in training datasets for foundational LLMs, including the ROOTS corpus [Laurençon et al., 2023] for training BLOOM [Scao et al., 2022] and The Pile [Gao et al., 2020a] for training LLaMA [Touvron et al., 2023]. The code data represented in these datasets are unlabeled raw source code from GitHub. There is also a family of code-only datasets for training or fine-tuning coding-specific LLMs, including The Stack [Kocetkov et al., 2022], a 3TB corpus of permissively licensed source code, preceded by CodeParrot with 50GB of deduplicated source code [Tunstall et al., 2022]. These massive datasets are usually used to train CodeLLMs. However, labeled data are required for training and evaluating LLMs for coding tasks involving source code and natural language descriptions. CodeXGLUE is a benchmark dataset Lu et al. [2021] for 10 coding tasks that include 14 subsets, four of which are code-text pairs. Most of the code-text pairs in CodeXGLUE come from CodeSearchNet.

CodeSearchNet (CSN) has also been employed for pretraining LLMs, enabling supervised learning techniques to achieve state-of-the-art performance for models such as CodeT5+ [Wang et al., 2023] and UniXcoder [Guo et al., 2022]. A few code-text pair datasets set out to surpass CSN in size. CoDesc combines existing parallel datasets (CSN, DeepCom [Hu et al., 2020], CONCODE [Iyer et al., 2018a], and FunCom [LeClair et al., 2019]), and then refines the results from the superset, which yielded 4.2M Java data samples. PyMT5 [Clement

---

[1] https://github.com/FSoft-AI4Code/TheVault

et al., 2020] is a dataset with 7.7M Python code-text. However, both of these datasets each contains code for a single programming language. Notable datasets created from Stack Overflow [2] include the necessary code-text data for generating post titles [Gao et al., 2020b, Liu et al., 2022].

## 3 The Vault dataset

### 3.1 Overview

In The Vault, we leverage a subset of The Stack [Kocetkov et al., 2022], recognized as the most expansive publicly available, multilingual, permissive-licensed source code dataset weighing in at 3TB. From this large-scale dataset, The Vault transforms raw source code into a collection of high quality pairs of code and text. Our transformation pipeline is designed to efficiently extract data from source code, create text-code pairings, and remove noise, yielding three distinct output datasets, as detailed in Figure 2. We draw from a subset of The Stack, which comprises code in 10 prevalent programming languages, such as C, C#, C++, Java, JavaScript, GoLang, PHP, Python, Ruby, and Rust (out of the total 300 languages featured in The Stack). Each language-specific raw source code feeds into a custom-built tree-sitter[3] parser.

This parser is designed to extract functions, classes, methods, block code snippets, and their corresponding block or inline comments. The figure 1 illustrated a basic structure of a code file that contains multiple levels of code snippets. By applying a breadth-first search on the Abstract Syntax Tree (AST) of the root node, the parser is able to traverse down different node and leaf levels (class, function, and inline), result three separate datasets:

1. The first output dataset, referred to as $D_{\text{paired}}$, contains pairs of classes (node 1) and functions (node 3) with corresponding block comments that serve as docstrings (node 2). After the initial construction, this dataset proceeds through a pipeline that employs both *rule-based filters* and *neural-based filters* to remove noisy samples that fail to meet the criteria detailed in Section 3.2.

2. The second output dataset, denoted as $D_{\text{unimodal}}$, consists of standalone functions and classes, not paired with any docstring or comments, thereby forming a unimodal dataset.

3. The third and final dataset, $D_{\text{block}}$, includes pairs of arbitrary code blocks (node 4) and inline comments (node 5). To construct this set, we capture all inline comments. Each comment is paired with the preceding code block, tagged as the "previous context" (node 4a), and the following code block, "next context" (node 4b).

A large number of block comments adhere to widely accepted docstring formats (Appendix A.5), encompassing neatly organized details about the name (identifier) of the associated function or class, their parameters, arguments, and return types. We channel these block comments through docstring parsers, which we have developed and made publicly available, to extract this information as metadata for each sample in our dataset. We contend that this metadata could prove beneficial for downstream tasks, prompt settings, and other applications (Figure 8). Collectively, these three datasets ($D_{\text{block}}$, $D_{\text{unimodal}}$, and $D_{\text{paired}}$) constitute The Vault. Note that through the evaluation process, only $D_{\text{paired}}$ is used since its contains data that is suitable for training and comparison with other datasets.

### 3.2 Data Cleaning Pipeline

From preliminary survey of the output dataset containing pairs of classes and functions with their corresponding block comments $D_{paired}$, we observe salient patterns that would impair the training quality for code related tasks. We implemented a set of rule-based filters (Section 3.2.1) to remove irrelevant information or reformat textual data to be more descriptive of the corresponding code block. To address cases where the code-text pairs have inadequate or erroneous semantic correlation, we trained a neural-based model based on CodeBERT (Section 3.2.2) to serve as a filter. Such a filter generates a score, which is used to assess the alignment of a pair of code and text. Low-scoring samples are assumed to be unaligned and will be removed.

### 3.2.1 Remove Noisy Sample by Rules

Our data pipeline employs 13 rule-based filters to eliminate noisy patterns in the source dataset. These filters, detailed in Table 1, are categorized into three main groups: enhancing readability, promoting consistency, and preserving the intended usage of the code.

---

[2] https://stackoverflow.com/
[3] https://tree-sitter.github.io/tree-sitter/

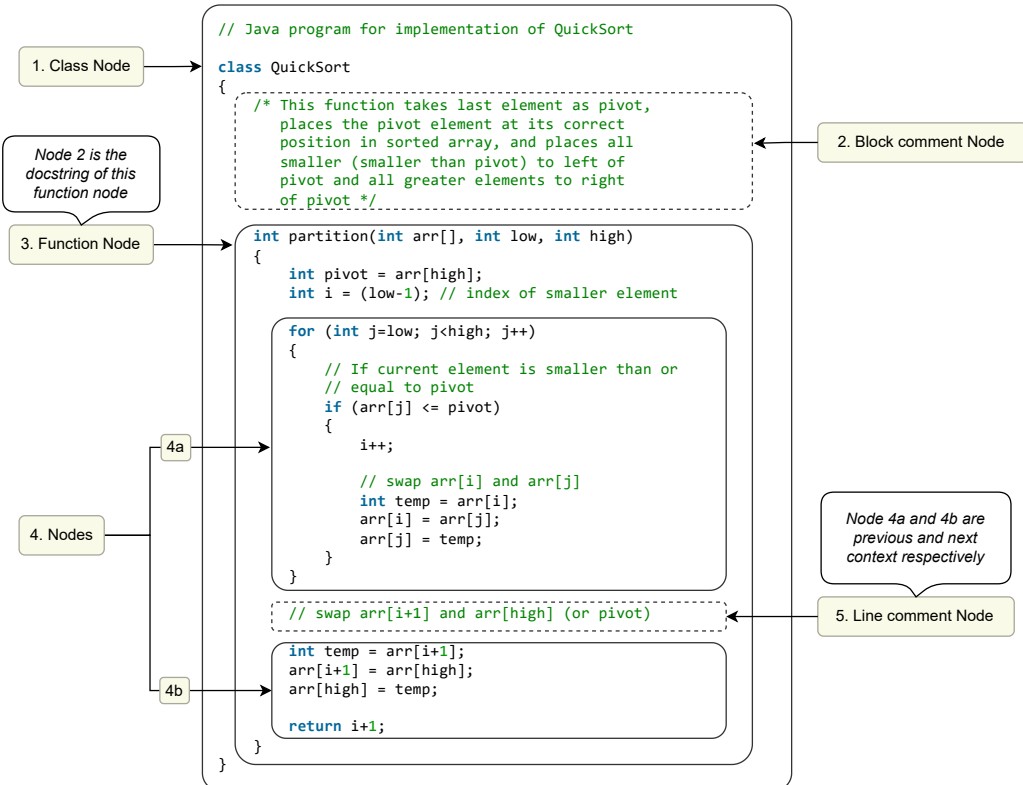

Figure 1: The tree-sitter node structure. Classes (1) and functions (3) are extracted along with their corresponding docstring, which may be in the form of a block comment (2) or a line comment (5). The line comments (5) are extracted along with their preceding (4a) and succeeding (4b) code nodes for the inline dataset.

In terms of readability, we strip delimiters, math formulas, HTML tags, and metadata tags from the text. This ensures a cleaner and more coherent code-text pairing. For consistency, we remove elements that may cause irregularities in the dataset. This includes stripping hyperlinks and embedded code, and removing empty comments, overly short or long comments, non-English comments, auto-generated blocks, and work-in-progress comments. Lastly, to preserve the original purpose of the code, we remove comments that are questions or serve as examples or notes. This rigorous filtering process guarantees a high-quality dataset, improving the effectiveness of code-focused language models.

### 3.2.2 Remove Low-Quality Samples with Neural-based Classifier

Beyond the use of rule-based filtering methods, a crucial question arises: how do we ensure alignment between code and text? Random comments unrelated to the functionality of the code snippet can contaminate the dataset, necessitating the removal of such misaligned samples to guarantee quality. To address this issue, we constructed a classifier utilizing CodeBERT [Feng et al., 2020b], de-

| Categories | Percentage (%) |
|---|---|
| *Readability* | |
| Strip Delimiters | 13.430 |
| Strip Math Formulas | 0.021 |
| Strip HTML Tags | 3.180 |
| Strip Metadata Tags | 5.260 |
| *Consistency* | |
| Strip Hyperlink | 0.510 |
| Strip Embedded Code | 12.680 |
| Remove Empty Comments | 71.470 |
| Remove Comments Too Short / Long | 4.100 |
| Remove Non-English Comments | 3.230 |
| Remove Auto-gen Blocks | 0.050 |
| Remove Work-in-Progress Comments | 0.002 |
| *Intended usage* | |
| Remove Comments as Questions | 0.020 |
| Remove Comments as Examples or Notes | 0.460 |

Table 1: The percentage of constructed code-text pairs from The Stack caught by each rule-based filter.

signed to score the semantic relationship between a function or class and its corresponding docstring.

In our scoring model, we input code snippets and docstrings separated by a token $$. Approximately 12% of the already rule-filtered code-text pairs dataset was randomly selected for training.

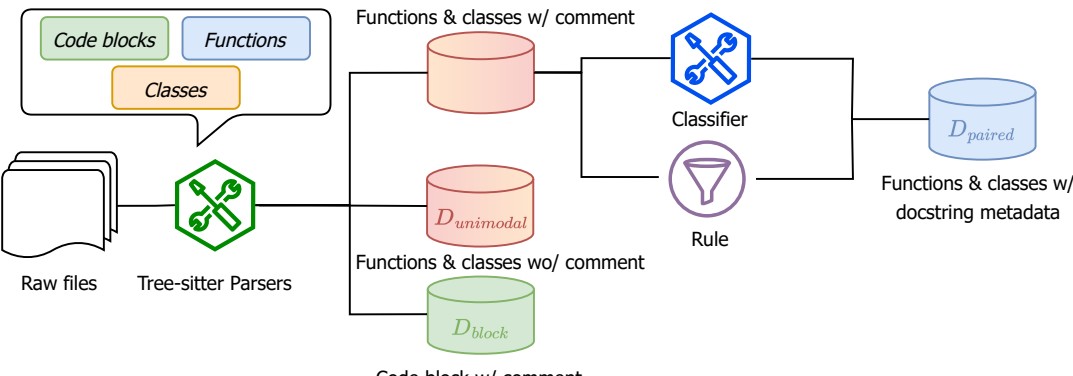

Figure 2: Pipeline to create datasets of code blocks with comments $D_{block}$, unimodal code $D_{unimodal}$, and code-text pairs $D_{paired}$ from raw source code.

| Language | Number of functions | | #Repositories | #Tokens | | |
|---|---|---|---|---|---|---|
| | w/docstring | All | | #Unique code token | #Unique docstring token | #Unique identifier |
| Python | 7,825,291 | 39,221,539 | 628,069 | 22,050,020 | 1,633,062 | 3,423,694 |
| PHP | 4,696,756 | 30,323,578 | 439,514 | 11,203,393 | 715,546 | 1,133,437 |
| JavaScript | 1,683,568 | 33,015,657 | 355,761 | 4,895,923 | 501,750 | 753,399 |
| Java | 6,667,422 | 69,744,181 | 321,129 | 16,536,979 | 1,749,151 | 2,525,492 |
| C# | 3,350,316 | 35,736,746 | 150,657 | 5,485,063 | 409,220 | 1,233,383 |
| C++ | 1,709,448 | 28,684,400 | 116,897 | 5,630,067 | 678,063 | 1,155,241 |
| C | 1,685,966 | 13,762,988 | 88,556 | 5,764,837 | 750,146 | 1,197,164 |
| Go | 5,153,436 | 23,832,763 | 241,238 | 6,818,885 | 2,472,000 | 1,918,773 |
| Rust | 864,987 | 8,230,575 | 68,615 | 2,130,327 | 221,877 | 315,331 |
| Ruby | 461,585 | 4,342,191 | 61,804 | 1,436,713 | 146,237 | 213,005 |
| Total | 34,098,775 | 286,894,618 | 2,364,144 | 73,077,761 | 7,351,960 | 12,869,338 |

Table 2: The size of extracted function data in each programming language.

As labeled data was unavailable, we generated negative samples by randomly pairing functions and docstrings within the same programming language. We then passed the representation of the $$ token to a linear layer, which produced a semantic correlation score between 0.0 and 1.0. Code-text pairs were then filtered using a binary classification gate with a threshold of 0.5.

To validate our model, we employed GPT 3.5 for analogous predictions. A million predictions were generated from unseen instances, from which we selected 300 per language: 200 high-confidence instances (100 consistent and 100 inconsistent code-text predictions) and 100 low-confidence instances. GPT 3.5-turbo was instructed to assign a consistency score (1-10) for each instance's code-docstring pair, serving as a benchmark for our model's predictions. For high-confidence instances, our model agreed with the GPT 3.5-turbo scores over 80% of the time. Although our model faced challenges with ambiguous samples, the Area Un-

der the Curve (AUC) metric proved suitable due to our primary goal of excluding misalignments while preserving matched examples. An average AUC of 0.89 indicates that our approach effectively reduced dataset noise without discarding numerous informative samples. Detailed configurations and evaluation results are available in Appendix A.2.

In addition, we use our model to find noisy examples in the rule-based noise-remove version of CodeSearchNet in CodeXGlue. Table 3 presents some inconsistent examples found by our model for Python, Java, JavaScript, and PHP in CSN. It can be observed that detected pairs show strong inconsistency between docstring and code. For instance, the docstring of the example in Python does not give much insight into what the code does or its purpose. The code defines a method named 'has_url' which checks if the attributes have a non-empty value; however, the docstring mentions templates which does not provide enough context to fully understand how this code relates to templates or

| Languages | Inconsistent pairs |
|---|---|
| Python | ```
// Handy for templates.
def has_urls(self):
    if self.isbn_uk or self.isbn_us or self.official_url or self.notes_url:
        return True
    else:
        return False
``` |
| Java | ```
// only for change appenders
public MapContentType getMapContentType(ContainerType containerType){
        JaversType keyType = getJaversType(Integer.class);
        JaversType valueType = getJaversType(containerType.getItemType());
        return new MapContentType(keyType, valueType);
    }
``` |
| JavaScript | ```
// we do not need Buffer pollyfill for now
function(str){
  var ret = new Array(str.length), len = str.length;
  while(len--) ret[len] = str.charCodeAt(len);
  return Uint8Array.from(ret);
}
``` |
| PHP | ```
// disini mo ba atur akan apa mo kamana
private function _parse_routes()
    {
        $uri=implode('/', $this->uri->segments());

        if (isset($this->router[$uri])) {
                return $this->_set_request(explode('/', $this->router[$uri]));
        }

        ...
    }
``` |

Table 3: Examples of Inconsistent pairs in CodeSearchNet found by our model in Python, Java, Javascript and PHP. "//" represents for docstring section. More examples are demonstrated in Table 15 in Appendix section.

| Dataset | #PL | #Function w/ docstring | w/o docstring |
|---|---|---|---|
| PyMT5 [Clement et al., 2020] | 1 | ≈ 7,700,000 | - |
| CoDesc [Hasan et al., 2021] | 1 | 4,211,516 | - |
| CodeSearchNet [Husain et al., 2019] | 6 | 2,326,976 | 4,125,470 |
| CodeXGLUE CSN [Lu et al., 2021] | 6 | 1,005,474 | - |
| Deepcom [Hu et al., 2020] | 1 | 424,028 | - |
| CONCODE [Iyer et al., 2018b] | 1 | 2,184,310 | - |
| Funcom [LeClair et al., 2019] | 1 | 2,149,121 | - |
| CodeT5 [Wang et al., 2021] | 8 | 3,158,313 | 5,189,321 |
| THEVAULT | 10 | 34,098,775 | 205,151,985 |

Table 4: Comparison of THEVAULT function set to other code-text datasets.

| Language | Training set | | | Valid set | Test set |
|---|---|---|---|---|---|
| | Small | Medium | Full | | |
| Python | 370,657 | 1,952,110 | 7,772,647 | 30,992 | 21,652 |
| Java | 351,213 | 1,612,366 | 6,629,193 | 22,677 | 15,552 |
| JavaScript | 82,931 | 404,729 | 1,640,416 | 22,044 | 21,108 |
| PHP | 236,638 | 1,155,476 | 4,656,371 | 21,375 | 19,010 |
| C | 105,978 | 381,207 | 1,639,319 | 27,525 | 19,122 |
| C# | 141,090 | 783,166 | 3,305,891 | 24,787 | 19,638 |
| C++ | 87,420 | 410,907 | 1,671,268 | 20,011 | 18,169 |
| Go | 267,535 | 1,319,547 | 5,109,020 | 19,102 | 25,314 |
| Ruby | 23,921 | 112,574 | 424,339 | 17,338 | 19,908 |
| Rust | 35,367 | 224,015 | 825,130 | 16,716 | 23,141 |
| Total | 1,702,750 | 8,356,097 | 33,673,594 | 222,567 | 202,614 |

Table 5: The proportion of training, validation, and test set of THEVAULT.

its broader purpose. Besides, our model is able to identify non-English samples, which are presented in the example of PHP, that are not captured by the rule-based methods.

## 4 Empirical Evaluation

In this section, we aim to assess the quality of The Vault in comparison with other datasets, such as CSN. To substantiate this quality, we fine-tune prominent CodeLLMs on tasks that necessitate the involvement of both code and text, including code summarization, code search, and code generation. We then compare these models, which have been fine-tuned on The Vault, with those fine-tuned on CSN. The comparison is made using the same test datasets and commonly employed metrics, such as MRR, smoothed BLEU [Lin and Och, 2004], and pass@k [Chen et al., 2021].

### 4.1 Dataset Statistics

Table 2 provides the statistics of the samples for each programming language after undergoing our data-cleaning pipeline. In total, we have approximately 34M samples. The table also includes other information, like the number of tokens for code and docstrings, and the quantity of repositories.

Table 4 offers a comparison between The Vault and other parallel datasets frequently used for pre-training and fine-tuning downstream tasks. These

| Model | Dataset | Python | Java | JavaScript | Go | PHP | Ruby | Total/Avg |
|---|---|---|---|---|---|---|---|---|
| | | CODESEARCHNET TESTSET (BLEU-4) | | | | | | |
| CodeT5 | raw/TheStack | 16.18 | 9.06 | 6.23 | 19.05 | 7.07 | 5.78 | 11.84/10.56 |
| | CodeSearchNet | 19.55 | 20.38 | 16.15 | 19.83 | 26.26 | 15.38 | **21.24/19.59** |
| | TheVault/small | 18.94 | 17.72 | 13.96 | 19.92 | 20.43 | 15.22 | 18.83/17.70 |
| PLBART | raw/TheStack | 0.86 | 3.06 | 0.59 | 10.91 | 2.29 | 0.47 | 3.23/3.03 |
| | CodeSearchNet | 17.99 | 17.38 | 14.84 | 17.98 | 22.54 | 14.08 | **18.78/17.47** |
| | TheVault/small | 14.93 | 15.66 | 11.95 | 17.03 | 18.00 | 11.49 | 15.95/14.84 |
| | | THEVAULT TESTSET (BLEU-4) | | | | | | |
| CodeT5 | raw/TheStack | 16.18 | 9.06 | 6.23 | 19.05 | 7.07 | 5.78 | 11.84/10.56 |
| | CodeSearchNet | 10.86 | 8.00 | 8.42 | 17.87 | 17.85 | 10.26 | 16.11/12.21 |
| | TheVault/small | 12.26 | 11.13 | 9.68 | 31.64 | 38.86 | 11.23 | **25.12/19.13** |
| PLBART | raw/TheStack | 1.69 | 4.02 | 0.43 | 24.60 | 4.83 | 0.49 | 7.19/6.01 |
| | CodeSearchNet | 10.24 | 7.26 | 7.64 | 16.90 | 13.83 | 9.60 | 14.39/10.91 |
| | TheVault/small | 10.23 | 9.28 | 8.95 | 22.78 | 34.32 | 9.74 | **20.29/15.88** |

Table 6: Smoothed BLEU-4 results for code summarization. The "Total" column demonstrates combined data in all languages to calculate BLEU, while "Avg" is the average BLEU score on the language level.

datasets include Funcom [LeClair and McMillan, 2019], Deepcom [Hu et al., 2020], CONCODE [Iyer et al., 2018b], CSN [Husain et al., 2019], CoDesc [Hasan et al., 2021], and non-public data used for pretraining [Clement et al., 2020, Ciurumelea et al., 2020, Wang et al., 2021].

We split the training set into two smaller subsets: the small set and the medium set that contain 5% and 20% of the full training set, respectively. To reduce data leakage during training, we employed the MinHash LSH technique [Zhu et al., 2023] to filter training instance clusters that are close to samples in the validation and test sets of CSN, HumanEval, and MBPP. Additionally, during dataset partitioning, we prevented content from the same repository from appearing in multiple sets, thereby avoiding any potential internal data leakage. A more detailed analysis of The Vault at the class and code block levels can be found in Appendix A.4.

### 4.2 Experiment Setup

**Data splitting:** During the experiment phase, The Vault ($D_{paired}$) was split into three distinct datasets: training, validating, and testing sets. To avoid data leakage, we reinforced a policy where code samples from the same repository must all be in the same set. In the splitting algorithm, we also included as a goal the preservation of the token length distribution from The Vault's dataset in each subset.

For richer comparisons, the training set was further branched off to two smaller sets, the small and medium training sets, sampling 5% and 20% of the full training set, respectively. Details about

experiment data can be found in Table 5. Note that TheVault/small has a comparable size with CSN, making it fair to assess and compare the quality of these two datasets.

Besides, in order to validate the efficiency of our processing pipeline, we conduct a comparison between the performance of models trained on The Stack (raw data) and The Vault (processed data). Specifically, we established three function-level subsets, each approximately the size of TheVault/small ($\approx$1.7M code-text instances). These subsets were created by randomly sampling the raw function-level dataset extracted from The Stack, without applying any filtering (referred to as raw/TheStack). We use three different seeds to sample raw/TheStack and report the average result. All experiments are conducted using 4 NVIDIA A100 GPUs.

**Code search:** We select CodeBERT [Feng et al., 2020a], RoBERTa [Liu et al., 2019] and UniX-Coder [Guo et al., 2022] as the encoder for embedding source code and natural language query. We train each model for 10 epochs with a sequence max length of 512, and a learning rate of $2^{-5}$.

**Code summarization:** CodeT5 [Wang et al., 2021] and PLBART [Ahmad et al., 2021a] are employed for the summarization task. We use the base versions and set the max input tokens to 512 and the max output tokens to 400. We train for 5 epochs with batch size of 512 and a learning rate of $2^{-4}$.

| Model | Fine-tune data | Python | Java | JavaScript | Go | PHP | Ruby | Avg |
|---|---|---|---|---|---|---|---|---|
| | | CODESEARCHNET TESTSET (MRR) | | | | | | |
| CodeBERT | raw/TheStack | 0.3713 | 0.3492 | 0.3148 | 0.5519 | 0.2731 | 0.2748 | 0.3559 |
| | CodeSearchNet | 0.3793 | 0.4636 | 0.4437 | 0.6201 | 0.4741 | 0.5219 | 0.4838 |
| | TheVault/small | **0.4074** | **0.4857** | **0.4466** | **0.6578** | **0.6578** | **0.5251** | **0.5301** |
| RoBERTa | CodeSearchNet | 0.3479 | 0.448 | 0.4254 | 0.5684 | 0.4623 | 0.5147 | 0.4611 |
| | TheVault/small | **0.4849** | **0.5581** | **0.4962** | **0.7446** | **0.5166** | **0.59** | **0.5651** |
| UniXCoder | CodeSearchNet | 0.3935 | 0.4549 | 0.4459 | 0.5861 | 0.489 | 0.5446 | 0.4857 |
| | TheVault/small | **0.4427** | **0.4909** | **0.4506** | **0.6416** | **0.4515** | **0.5702** | **0.5079** |
| | | THEVAULT TESTSET (MRR) | | | | | | |
| CodeBERT | raw/TheStack | 0.318 | 0.3245 | 0.1837 | 0.4194 | 0.1718 | 0.0878 | 0.2509 |
| | CodeSearchNet | 0.2881 | 0.3213 | 0.2409 | 0.4123 | 0.1854 | 0.2579 | 0.2843 |
| | TheVault/small | **0.3501** | **0.4214** | **0.3216** | **0.4864** | **0.2351** | **0.2904** | **0.3165** |
| RoBERTa | CodeSearchNet | 0.2644 | 0.3329 | 0.2371 | 0.2375 | 0.1577 | 0.2574 | 0.2478 |
| | TheVault/small | **0.4533** | **0.5519** | **0.4386** | **0.5021** | **0.2876** | **0.3717** | **0.4342** |
| UniXCoder | CodeSearchNet | 0.2959 | 0.344 | 0.2508 | 0.185 | 0.1646 | 0.2669 | 0.2512 |
| | TheVault/small | **0.3852** | **0.4279** | **0.3491** | **0.4628** | **0.238** | **0.3201** | **0.3639** |

Table 7: Comparison between the models fine-tuned on the CODESEARCHNET and on different THEVAULT training subsets on code search task.

**Code generation:** We use CodeGen 350M and 2B Multi [Nijkamp et al., 2023] to evaluate code generation. We use the same configuration as in the code summarization task.

### 4.3 Evaluation Results

#### 4.3.1 Code Summarization

For this task, we utilize the Vault and CSN to fine-tune CodeT5 and PLBART to summarize the source code. The Vault and CSN exhibit significant differences in docstring format. The Vault retains the complete docstring format, offering comprehensive descriptions of core logic, parameters, arguments, and return types. This feature enables versatile applications in code documentation and various downstream tasks. Additionally, we save the first sentence of each complete docstring as metadata, termed as *short_docstring*. To facilitate fair comparison between The Vault and CSN, we apply post-processing to our full docstrings and *short_docstrings* training sets, thereby reducing format distribution disparity.

Table 6 shows the results when comparing CodeT5 and PLBART trained on CSN and The Vault for the code summarization task, we report the best score when using full docstrings and *short_docstrings*. We present further experimental outcomes using the Rouge-L [Lin, 2004] and BERTScore [Zhang et al., 2020] metrics in

Appendix, Table 14. The results show that our pipeline has witnessed strong effectiveness compared to unprocessed data, raw/TheStack. Particularly, during training on the raw/TheStack dataset for the code summarization task, we found that the PLBART and CodeT5 generate outputs with substantial noise. These outputs are characterized by a prevalence of special tokens like "//" and "*". This finding strongly underscores the efficacy of our filtering process in enhancing the quality of the dataset. However, the result using CSN shows superior performance on CSN's testset than using The Vault. The reason for this is our mention of the post-processing step to reduce the difference between the CSN and The Vault filtering methods, where the syntactic distribution can still exhibit nonidentical characteristics, which can affect the BLEU score. However, this gap could be reduced by using the full version of The Vault as shown in Table 14. Although the total performance gain when evaluated on the CSN test set is marginal (21.73 versus 21.24), it is worth noting that, despite the intermediary processing, CSN is a considerably smaller dataset with more consistent docstring patterns. In contrast, our dataset is substantially larger and exhibits greater diversity, thereby encouraging broader generalization. When evaluated against The Vault's test set, the model fine-tuned on CSN lags behind by over 10%.

| Model | Fine-tune dataset | pass@1 | pass@10 | pass@100 |
|---|---|---|---|---|
| | HUMANEVAL | | | |
| CodeGen 350M | - | 6.67 | 10.61 | 16.84 |
| | Py/CodeSearchNet | 2.76 | 8.76 | 14.72 |
| | (250K) Py/TheVault | 3.74 | 10.57 | 16.26 |
| | raw/PyTheStack | 6.64 | 15.42 | 24.80 |
| | Py/TheVault | **8.14** | **18.12** | **30.07** |
| CodeGen 2B | - | **14.51** | 24.67 | 38.56 |
| | Py/TheVault | 14.00 | **25.74** | **41.72** |
| | MBPP | | | |
| CodeGen 350M | - | 7.46 | 24.18 | 46.37 |
| | Py/TheVault | **10.13** | **33.96** | **53.20** |
| CodeGen 2B | - | 18.06 | 45.80 | **65.34** |
| | Py/TheVault | **27.82** | **50.06** | 65.06 |

Table 8: Result on code generation benchmarks using CodeGen Multi 350M and 2B models.

### 4.3.2 Code Search

We utilize CodeBERT, RoBERTa and UniXCoder to fine-tune both The Vault and CSN for the purpose of the code search task. We also furnish a baseline Mean Reciprocal Rank (MRR) score. MRR is a widely used metric for evaluating code search tasks, and in our case, it is trained on 10 different programming languages and assessed using the test set from CSN and The Vault. The results of this task, when fine-tuning the model on The Vault and CSN, are illustrated in Table 7. Remarkably, we attain superior results in most languages when fine-tund using the smallest dataset, TheVault/small, in contrast to solely fine-tuning on the CSN corpus. Surprisingly, RoBERTa, a model pretrained on natural language text, outperforms the two code-pretrained models when evaluated on code search. This could imply the importance of natural language text representation over code representation in this task. Furthermore, models trained on The Vault consistently outperform all baseline models trained on raw/TheStack, underscoring both the efficiency of our processing pipeline and the dataset's ability to generalize across different architectures.

### 4.4 Code Generation

We experiment with two versions of CodeGen Multi [Nijkamp et al., 2023], which are 350M and 2B models on the HumanEval and MBPP benchmarks for code generation. The scope of our experiment was limited because the benchmarks only support Python. We use these checkpoints and continue fine-tuning them on The Vault because CodeGen Multi models are trained on the dataset with multiple languages.

To create Py/CodeSearchNet and Py/TheVault, we use the Python subsets of CSN and TheVault, respectively. We sampled the training Python set of

TheVault to match the size of the Python subset in CSN with 250K samples in the first round of fine-tuning. Additionally, raw/PyTheStack is a subset of Python data from The Stack mirroring the size of Python data present in The Vault dataset, which helps us to demonstrate the advancements achieved in our data process pipeline.

The results are shown in Table 8. We can see that fine-tuning the CodeGen Multi 350M on The Vault causes the model to improve significantly in terms of pass@1, pass@10, and pass@100 on the HumanEval and MBPP benchmarks. Additionally, CodeGen 2B is used to assess The Vault on larger scale models. Similar to experiments on small models, Table 8 shows that The Vault can improve the performance of pretrained large-scale models. These results validate The Vault's ability to improve the performance of pre-existing pretrained models. In the future, we will expand our evaluation to even larger scale models and assess The Vault's impact on them.

## 5 Conclusion

In this paper, we presented The Vault, a large dataset of high-quality code-text pairs from ten programming languages, with over 43 million samples. The Vault was carefully curated to ensure that each pair meets quality standards, with detailed and informative descriptions and consistent coding styles. Our analysis uncovered a number of intriguing patterns and trends that shed light on the characteristics of programming languages and coding practices. We believe that The Vault will be a valuable resource for researchers and practitioners in this rapidly evolving field, providing a solid foundation for developing novel approaches and advancing state-of-the-art code large language models.

## Limitations

In our approach, we employed 13 heuristic and context-specific rule-based filters, curated from manual data observations. While these filters effectively mitigated noisy patterns, their deterministic nature precluded comprehensive generalizability. To address this, we supplemented these rules with a neural-based approach as described in Section 3.2.2. However, the absence of labeled training data necessitated pseudo-random sample generation, which could compromise model soundness and potentially eliminate quality code-text pairs. Although cross-validation with GPT 3.5-turbo occasionally revealed scoring inconsistencies, we believe that human labeling and model fine-tuning could further refine the dataset.

Compared to The Stack and The Pile, our dataset is smaller, mainly due to our rigorous quality control procedures. Moreover, creating AST parsers for each programming language is a non-trivial task, limiting our dataset to 10 popular programming languages compared to The Stack's 300. Nonetheless, our framework's codebase is publicly available, encouraging future contributions to extend our parsers and rules to additional languages.

The current study primarily utilized small models with less than 2 billion parameters to illustrate the value of The Vault. These models effectively demonstrated the dataset's potential, but further research with larger models would shed light on its robustness and scalability across more complex tasks. In future work, we plan to conduct experiments using large-scale language models to further assess the impact of our dataset.

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

# A Appendix

## A.1 Rule-based filters

While some datasets eliminate all special characters (!@#$%&*()_-+=/.,'— ') and keep only the first sentence or the paragraph preceding the first double endline symbol [Hasan et al., 2021, Mahmud et al., 2021], our heuristic rules take a different approach. Instead of discarding such characters outright, we selectively remove the noisy elements while aiming to capture as many informative sections as possible.

We analyze each docstring block individually and retain the sections that meet our quality criteria. Table 9 provides comprehensive descriptions of our 13 rule-based filters, accompanied by illustrative examples. Additionally, table 10 presents the corresponding percentages of code-text pairs generated through the application of these rule-based filters.

## A.2 Neural-based refinement method

To detect semantic inconsistency between code-text pairs, we considered fine-tuning on large foundational models such as CodeGen [Nijkamp et al., 2023], BLOOM [Scao et al., 2022] or leverage GPT 3.5-turbo APIs. However, these approaches would incur very high costs in terms of financial resources, time, and computational power. We decided to train a dedicated model to deal with this specific task and use GPT 3.5-turbo to cross-check the predictions.

**Training:** We trained our model based on Code-BERT, [Feng et al., 2020a]. The model assigns a score for semantic correspondence between code and text, before passing through binary classification into Consistent and Inconsistent categories. We randomly chose 5M samples (500K for each language in The Vault) and divided them into training, validation, and testing sets at a ratio of 3:1:1. The input to the model is the concatenation of the docstring and the code together with the $$ token used to separate them (Figure 3). We use the representation of the $$ token and feed it into a linear layer to obtain the output logit.

Since labeled data was unavailable, we utilized self-supervised learning. We created negative samples by randomly pairing a function with a docstring from the same programming language (Figure 3).

**Cross-check:** We used GPT 3.5-turbo to perform similar classifications for semantic consistency of code-text pairs. We used a prompting template to ask GPT 3.5-turbo to score each pair of code-text on a scale of 1 to 10 for semantic correspondence with a detailed explanation and ran this prompting template on systematically selected 300 data points from each language with 100 data points in each of the following groups:

- Consistency group: Examples that the model gives high confidence prediction to class Consistent. We select the top 100 based on the output probability for class 1.

- Inconsistency group: Examples that the model gives high confidence prediction to class Inconsistent. We select the top 100 based on the output probability for class 0.

- Uncertainty group: Examples that the model gives uncertain predictions. We select the lowest top 50 examples for each class.

The systematic sampling scheme helped us select 2994 samples in function level to be scored out of millions, reducing the cost of requesting GPT 3.5-turbo API while enabling meaningful analysis. The prompt input to GPT 3.5-turbo is as follow:

```
I want you to act as an unbiased
    docstring evaluator for code. I will
     give you a docstring along with a
    source code, and you will give me a
    score for the consistency between
    them. The score will be on a scale
    of 1 to 10, 10 means the docstring
    can effectively summarize the code
    while 1 means they are inconsistent.
     The response answers must contain
    the score and the explanation that
    follows the format in the response
    format.

- Response format:
Score: X
Explanation: Y

- Docstring:
"{docstring}"

- Code:
"{code}"
```

**Empirical Evaluation Results:** Table 11 presents the performance of our model with GPT 3.5 turbo's scores as a reference, along with the scoring result for each group. In groups with high confidence, we witness a strong correlation between our model and GPT 3.5-turbo, with a high score for Consistency (7.81) and a low score for Inconsistency (3.15). A similar pattern is observed in the Uncertainty group, where the average score is close to the middle of the scale at 5.74.

| Categories | Syntax Feature | Action | Docstring |
|---|---|---|---|
| Comment Delimiter | Unnecessary comment delimiter | Update | `/**`
`* Lexical essentially tokenizer.`
`*`
`*/`

→ `Lexical essentially tokenizer.` |
| Hyperlink | URL Link | Update | `Deletes a Mux asset`
`@see`
`    https://docs.mux.com/v1/reference#deletean-asset`

→ `Deletes a Mux asset` |
| Embedded Code | Inline or embedded code snippets, command lines, or script excerpts | Update | `Set the trust level for a key in GPG keychain.`
`code-block:: bash`
`salt '*' gpg.trust-key key-id='3FAD9F1E'`
`trust-level='marginally'`

→ `Set the trust level for a key in GPG keychain.`
`code-block::  bash` |
| Question | Question: Why? How?, … | Update | `isup <url> – Is it down for everyone, or just you?`

→ `isup <url>` |
| Math formula | \sqrt(), \exp(), \mathbf, … | Update | `Recursive filter design using a least-squares`
`    method.`
`{[}B,A{]} = YULEWALK(N,F,M) finds the N-th order`
`recursive filter coefficients B and A.`

→ `Recursive filter design using a least-squares`
`method.` |
| Metadata Tag | Metadata tags or annotations | Update | `Creates a slice of 'array' with 'n' elements`
`    dropped`
`from the end.`
`@static`
`@memberOf_`
`@since 3.0.0`

→ `Creates a slice of 'array' with 'n' elements`
`dropped from the end.` |
| HTML Tags | HTML tags: <p>... </p>, … Special tags. | Update | `Constructs a GeneralStoresProductModel`
`from a plain JavaScript object.`

→ `Constructs a GeneralStoresProductModel from a`
`plain JavaScript object.` |
| Example and note | Code example, note from developers | Update | `Pull packages data dir.`
`note:  Uses su to access package's data dir.`

→ `Pull packages data dir.` |
| Unsuitable Length | Length < 5, length > 500 | Remove | `Write objects` |
| Non-English | Not written in English | Remove | `Retorna uma estrutura com os argumentos`
`passados para o programa.` |
| Auto-gen | Auto-generated | Remove | `*<!-begin-user-doc->`
`<!-end-user-doc->`
`@generated` |
| Under-dev | Under-development | Remove | `Deprecate this build, so that it will be rebuilt if`
`any other test run wants to use it.` |
| No comment | No docstring/comment in function | Remove | `null` |

Table 9: Rule-based filters and examples.

| Categories | Python | PHP | JavaScript | Java | C# | C++ | C | Rust | Ruby | Go | Total |
|---|---|---|---|---|---|---|---|---|---|---|---|
| Comment Delimiter | 12.02 | 33.38 | 9.94 | 11.98 | 16.7 | 6.92 | 13.28 | 8.43 | 9.13 | 4.95 | 13.43 |
| Hyperlink | 0.95 | 0.44 | 0.66 | 0.25 | 0.71 | 0.15 | 0.11 | 0.59 | 1.11 | 0.65 | 0.51 |
| Embedded Code | 31.65 | 1.09 | 1.38 | 1.41 | 1.39 | 6.51 | 6.16 | 0.67 | 3.18 | 2.41 | 12.68 |
| Question | 0.03 | 0 | 0.02 | 0.02 | 0.01 | 0.03 | 0.02 | 0.06 | 0.13 | 0.02 | 0.02 |
| Math formula | 0.1 | 0 | 0.01 | 0.01 | 0.01 | 0.02 | 0.02 | 0.01 | 0 | 0 | 0.021 |
| Metadata Tag | 0.62 | 6.81 | 1.86 | 2.69 | 2.15 | 4.35 | 6.14 | 0.83 | 1.69 | 0.46 | 5.26 |
| HTML Tags | 0.79 | 0.68 | 0.8 | 2.7 | 17.15 | 0.31 | 0.45 | 1.13 | 1.56 | 0.13 | 3.18 |
| Example and note | 1.4 | 0.26 | 0.36 | 0.34 | 0.22 | 0.18 | 0.4 | 0.45 | 0.79 | 0.3 | 0.46 |
| Unsuitable Length | 5.11 | 8.79 | 3.90 | 2.20 | 2.75 | 4.58 | 3.86 | 2.26 | 5.19 | 4.37 | 4.10 |
| Non-English | 1.69 | 5.72 | 3.26 | 4.16 | 2.62 | 4.1 | 1.94 | 0.42 | 1.53 | 1.77 | 3.23 |
| Auto-gen | 0.01 | 0 | 0 | 0.2 | 0 | 0 | 0 | 0 | 0 | 0 | 0.05 |
| Under-dev | 0.02 | 0 | 0 | 0 | 0 | 0 | 0 | 0 | 0 | 0 | 0.002 |
| No comment | 60.54 | 49.0 | 78.5 | 77.15 | 76.16 | 80.95 | 72.28 | 80.43 | 71.55 | 69.75 | 71.47 |

Table 10: The percentage of constructed code-text pairs from The Stack caught by each rule-based filter, by programming language.

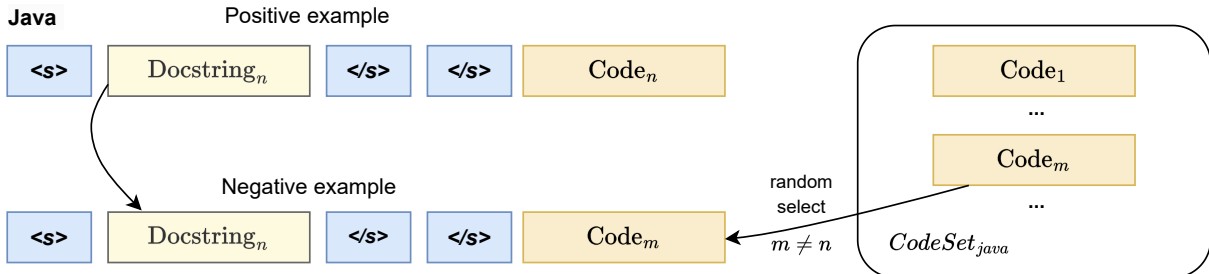

Figure 3: Input representation and Negative sample generation for code-docstring inconsistency detection.

In addition, we use GPT 3.5-turbo's scores to generate pseudo-labels and calculate accuracy and AUC for our model. We set a relative threshold of 5 to determine the labels. It can be witnessed that our model performs well in high-confidence groups but struggles in the uncertainty group. However, the accuracy is influenced by the choice of relative threshold, we consider Area Under the Curve (AUC) to measure the false positive and true positive rates. The metric shows a convincing result averaging 0.89, enabling us to effectively reduce a high amount of noise in our dataset while avoiding excluding too many informative examples. Finally, after removing noisy data using the proposed neural-based method, we notice a decrease of 1.3% in the dataset.

We use our model to find noisy examples in the rule-based noise-remove version of CodeSearchNet in CodeXGlue. Table 15 illustrates some examples found in 6 programming languages. It can be observed that detected pairs show strong inconsistency between docstring and code. For instance, the docstring of the first example in Python does not give much insight into what the code does or its pur-

pose. The code defines a method named 'has_url' which checks if the attributes have a non-empty value; however, the docstring mentions templates which does not provide enough context to fully understand how this code relates to templates or its broader purpose. A similar pattern also presents in the remaining examples. An example that provides more clarity is the second example in Ruby. The docstring describes a function with a 'YAML filePath' parameter, but the function itself does not actually have this parameter. Besides, our model is able to identify non-English samples (the second example in PHP) that are not captured by the rule-based method.

### A.3 Analysis of Function-Level Data in The Vault

Detailed description of function level data in The Vault can be found in Figure 4.

### A.3.1 Code and Docstring Analysis

**Token Length Distribution:** When training seq-to-seq LLMs, maximum input and output lengths are typically required. By understanding the distribution of sequence lengths in the corpus, we can

| Language | GPT 3.5-turbo score (accuracy) | | | Accuracy (%) | AUC |
|---|---|---|---|---|---|
| | Consistency | Inconsistency | Uncertainty | | |
| Python | 8.19 ± 1.15 (99%) | 3.76 ± 1.96 (69%) | 6.20 ± 2.12 (44%) | 70.67 | 0.8559 |
| PHP | 7.73 ± 1.32 (96%) | 3.01 ± 1.45 (90%) | 4.90 ± 2.23 (49%) | 78.33 | 0.8863 |
| JavaScript | 7.73 ± 1.25 (99%) | 2.95 ± 1.40 (89%) | 5.58 ± 2.29 (49%) | 79.00 | 0.8984 |
| Java | 7.65 ± 1.71 (94%) | 2.73 ± 1.32 (93%) | 5.83 ± 2.12 (53%) | 80.00 | 0.9014 |
| C# | 7.70 ± 1.35 (97%) | 3.31 ± 1.56 (82%) | 5.35 ± 2.09 (46%) | 75.00 | 0.8606 |
| C++ | 7.51 ± 1.64 (92%) | 2.82 ± 1.46 (89%) | 5.80 ± 2.33 (57%) | 79.33 | 0.8787 |
| C | 7.79 ± 1.10 (98%) | 2.99 ± 1.48 (88%) | 5.81 ± 2.08 (47%) | 77.67 | 0.9108 |
| Go | 8.08 ± 1.21 (99%) | 3.68 ± 1.67 (74%) | 6.09 ± 2.06 (50%) | 74.83 | 0.8819 |
| Rust | 8.03 ± 1.20 (99%) | 3.72 ± 1.77 (75%) | 6.83 ± 1.62 (50%) | 74.67 | 0.9051 |
| Ruby | 7.72 ± 1.03 (98%) | 2.51 ± 1.04 (96%) | 5.01 ± 2.23 (49%) | 81.00 | 0.9203 |
| All | 7.81 ± 1.33 (97%) | 3.15 ± 1.59 (84%) | 5.74 ± 2.19 (49%) | 77.05 | 0.8874 |

Table 11: Evaluate CodeBERT using the consistency score provided by GPT 3.5-turbo. We report the mean ± the standard deviation for the score in each subset.

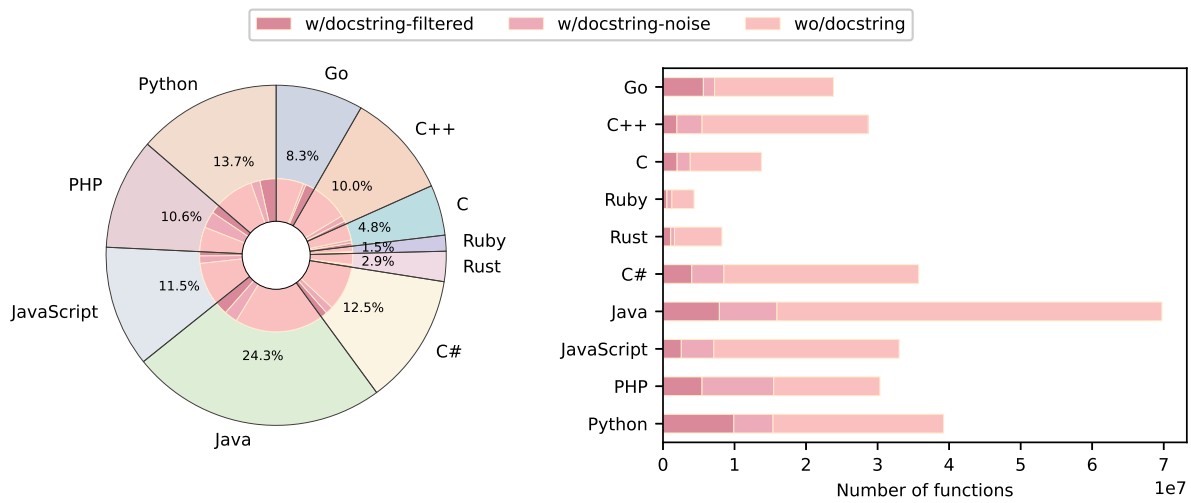

Figure 4: Distribution and the number of functions by the presence of docstrings. Functions with docstrings are further divided into two categories: functions removed by rule-based filters and functions in the final code-text dataset.

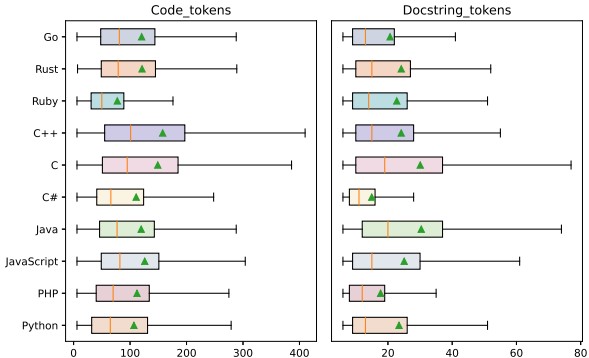

Figure 5: Code and Docstring tokens length distribution. The plot shows the lower to upper quartile values of the number of tokens in the data. The orange solid line indicates the median and the green triangle ▲ presents the mean.

choose appropriate input and output lengths for training. This can help improve the performance of training a language model and prevent the resulting LLMs from producing outcomes too short or too long for the intended use cases [Kaplan et al., 2020, Brown et al., 2020].

Our tokenization process utilizes the tree-sitter framework to parse source code into nodes on an abstract syntax tree; each node is considered a token. For docstring tokenization, we tokenize by word and punctuation. The code and docstring tokens length distribution for each programming language is illustrated in Figure 5. The number of tokens present in a function (average of around 100 tokens) is considerably more than the number of tokens found in the docstrings (average of 15-30 tokens) that describe it. In particular, among the

10 programming languages, C and C++ have the highest number of tokens in a function. This can be attributed to the fact that these languages are low-level languages, which typically require more code to perform a task when compared to higher-level languages. In the case of docstrings, their number of tokens is determined not only by the naturalness of the description in practice but also by cleaning rules outlined in Section 3.2.1. From Figure 5-Right and Table 10, it can be observed that the docstrings in Java and C are lengthy but are slightly cleaned by update-action rules, indicating that the docstrings in these two languages are typically long and more detailed in practice. Meanwhile, the number of tokens of docstrings in C# is the lowest. The cleaning rules may have played a role, as a significant proportion of the samples in C# has been updated based on *Comment Delimite* (16,7%) and *HTML Tags* (17,15%) rules.

Table 2 depicts the overall number of distinct tokens for each programming language. As our dataset contains extensive unique tokens, we believe that model training on The Vault can effectively handle unseen tokens. Besides, we find that multiple function names are reused due to the relatively small number of unique identifiers compared to the total number of functions in the dataset. This finding implies that even for humans, naming functions might be a difficult task.

**Docstring Styles**: Alongside typical docstrings that provide brief descriptions of the source code, many adhere to formatting and style conventions like Google, Jsdoc, and reST styles, among others. Our toolkit, designed to parse docstrings and extract metadata into a dictionary, supports 11 prevalent docstring styles. The styles we support and the information we aim to extract are depicted in figures 10 and 8 in Appendix A.5. This rich dataset could inspire research on advanced problems, such as controlling docstring style during generation or crafting explanations for function parameters.

Figure 9 provides statistics on the number of docstrings following a standard style. The data suggests that styled docstrings constitute a small fraction of the overall code-text dataset. One possible explanation is that our style detection rules are stringent, excluding docstrings with even minor syntax deviations, which might result in underestimating the number of docstrings adhering to a specific format. For styled docstrings, Figure 9-bottom presents the distribution of the number

of extracted attributes for each programming language, with most having between 1 to 5 elements. We make our docstring-style parser available to the community to facilitate easy customization and enhancement.

## A.4 Analyzing for Class and Inline Comment Set

In Table 12, we provide a statistical analysis of the number of classes and inline comments in both the raw set and the filtered set. Since the class structure is not defined in C and Go, we do not have their information to give in this table.

Initially, we excluded a substantial number of class samples from the raw dataset that lacked docstrings. The remaining class-docstring pairs underwent additional processing. Since the nature of classes and functions is similar, their functionalities can be meaningfully defined by pairs of a code snippet and a docstring. However, one of the problems when constructing paired data for class-comment samples is the large code snippet length of the class structure. As a result, we set the maximum number of code tokens that a class can have to 5000. On average, the code-token length of the class set is approximately 500, which is around five times longer compared to the average token length in the function set, while the number of docstring-token lengths is similar between the two sets, as shown in Figure 6. Each pair of class-docstring is also examined via a rule-based filtering process, as described in Section 3.2.1, serving as a sample point in $D_{pair}$ dataset.

In the $D_{block}$ analysis, we initiate the initial formation of the sub-dataset by identifying and extracting inline comments within code functions. The extracted comments undergo a series of cleaning procedures similar to those applied to the docstrings (as discussed in Section 3.2.1). After eliminating noisy samples, we proceed to establish various intervals for the number of comment tokens, aiming to determine the optimal upper and lower bounds that yield high-quality collected comments. Our observations reveal that inline comments exceeding 15 tokens typically incorporate code snippets, while comments containing fewer than 3 tokens lack substantial meaningful information. Consequently, this interval serves as a filtering criterion to generate the final version of $D_{block}$. Figure 7 shows the distribution of code-token length and docstring-token length in $D_{block}$ set.

| Language | Number of raw classes | | Number of classes after filtering | Number of raw inline comments | Number of inline comments after filtering |
|---|---|---|---|---|---|
| | w/ comment | wo/ comment | | | |
| Python | 497,550 | 1,440,539 | 422,187 | 24,066,884 | 14,013,238 |
| PHP | 2,223,472 | 6,232,180 | 1,173,916 | 9,892,486 | 5,873,744 |
| JavaScript | 494,819 | 2,409,932 | 291,479 | 4,426,086 | 1,438,110 |
| Java | 8,438,772 | 11,997,783 | 4,872,485 | 24,982,298 | 17,062,277 |
| C# | 2,378,379 | 9,097,968 | 1,437,800 | 10,130,704 | 6,274,389 |
| C++ | 285,184 | 791,355 | 174,370 | 20,770,494 | 10,343,650 |
| Rust | 188,517 | 3,591,465 | 93,311 | 2,998,368 | 2,063,784 |
| Ruby | 721,338 | 2,903,507 | 353,859 | 1,236,143 | 767,563 |
| C | - | - | - | 16,009,812 | 6,778,239 |
| Go | - | - | - | 7,574,542 | 4,390,342 |
| Total | 15,228,031 | 38,464,729 | 8,819,407 | 122,087,817 | 69,005,336 |

Table 12: The number of classes and inline comments associated with the class and inline set. The symbol '-' indicates that this information is unavailable due to the nonexistence of traditional classes in C and Go.

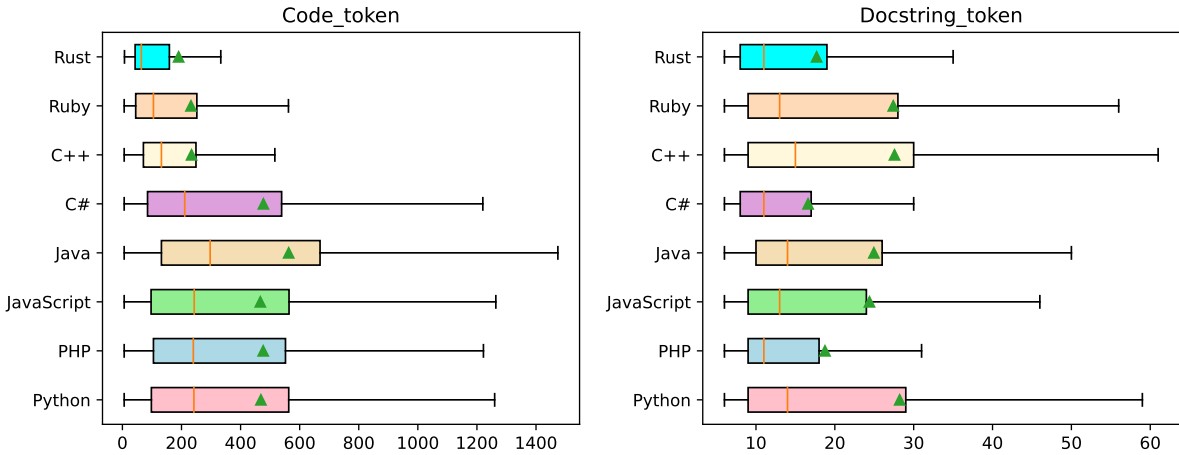

Figure 6: Code and Docstring tokens length distribution of the Class set after filtering.

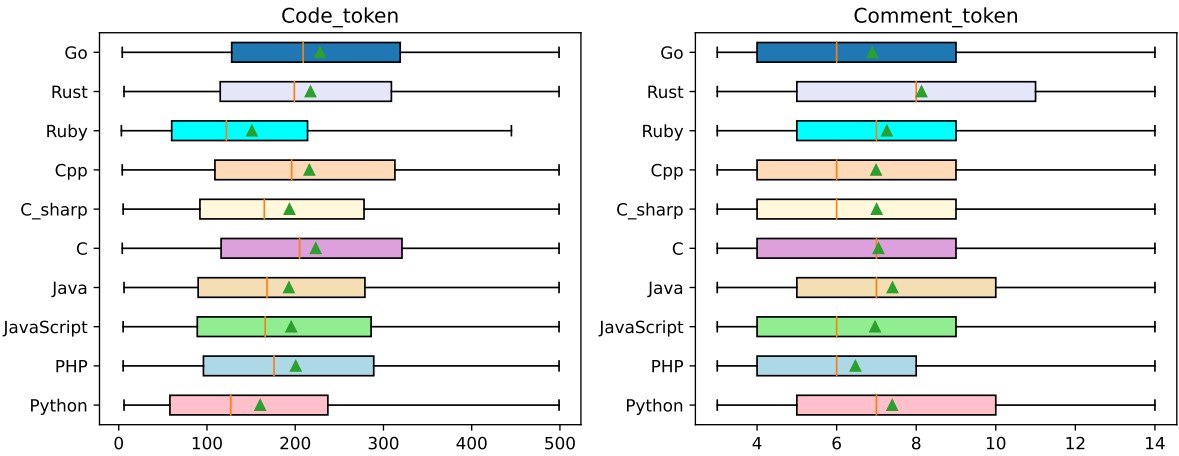

Figure 7: Code and Docstring tokens length distribution of $D_{block}$ set after filtering.

## A.5 Docstring Styling

A docstring is a string literal used as a form of documentation for a module, function, class, or method definition in programming languages. It is usually placed as the first statement in the code block (which can be inside or outside the code block itself) and enclosed by a comment delimiter (e.g.,

| Model | Fine-tune data | Python | Java | JavaScript | Go | PHP | Ruby | Rust | C | C++ | C# | Avg |
|---|---|---|---|---|---|---|---|---|---|---|---|---|
| | | CodeSearchNet Testset (MRR) | | | | | | | | | | |
| CodeBERT | CodeSearchNet | 0.3793 | 0.4636 | 0.4437 | 0.6201 | 0.4741 | 0.5219 | - | - | - | - | 0.4838 |
| | TheVault/small | 0.4074 | 0.4857 | 0.4466 | 0.6578 | 0.6578 | 0.5251 | - | - | - | - | 0.5301 |
| | TheVault/medium | 0.6585 | 0.6945 | 0.6197 | 0.8571 | 0.638 | 0.7096 | | - | - | - | 0.6962 |
| | TheVault | **0.6952** | **0.7242** | **0.6562** | **0.8789** | **0.6646** | **0.7474** | - | - | - | - | **0.7278** |
| RoBERTa | CodeSearchNet | 0.3479 | 0.448 | 0.4254 | 0.5684 | 0.4623 | 0.5147 | - | - | - | - | **0.6952** |
| | TheVault/small | **0.4849** | **0.5581** | **0.4962** | **0.7446** | **0.5166** | **0.59** | - | - | - | - | **0.5651** |
| UniXCoder | CodeSearchNet | 0.3935 | 0.4549 | 0.4459 | 0.5861 | 0.489 | 0.5446 | - | - | - | - | 0.4857 |
| | TheVault/small | **0.4427** | **0.4909** | **0.4506** | **0.6416** | **0.4515** | **0.5702** | - | - | - | - | **0.5079** |
| | | TheVault Testset (MRR) | | | | | | | | | | |
| CodeBERT | CodeSearchNet | 0.2881 | 0.3213 | 0.2409 | 0.4123 | 0.1854 | 0.2579 | - | - | - | - | 0.2843 |
| | TheVault/small | 0.3501 | 0.4214 | 0.3216 | 0.4864 | 0.2351 | 0.2904 | 0.326 | 0.2996 | 0.3015 | 0.3483 | 0.3165 |
| | TheVault/medium | 0.5929 | 0.6215 | 0.549 | 0.6862 | 0.3642 | 0.514 | 0.5705 | 0.5362 | 0.5264 | 0.5268 | 0.5488 |
| | TheVault | **0.6448** | **0.6633** | **0.592** | **0.7111** | **0.3891** | **0.5607** | **0.6243** | **0.5947** | **0.5932** | **0.5616** | **0.5935** |
| RoBERTa | CodeSearchNet | 0.2644 | 0.3329 | 0.2371 | 0.2375 | 0.1577 | 0.2574 | - | - | - | - | 0.2478 |
| | TheVault/small | **0.4533** | **0.5519** | **0.4386** | **0.5021** | **0.2876** | **0.3717** | **0.4195** | **0.3805** | **0.37** | **0.4099** | **0.4342** |
| UniXCoder | CodeSearchNet | 0.2959 | 0.344 | 0.2508 | 0.185 | 0.1646 | 0.2669 | - | - | - | - | 0.2512 |
| | TheVault/small | **0.3852** | **0.4279** | **0.3491** | **0.4628** | **0.238** | **0.3201** | **0.363** | **0.2934** | **0.2861** | **0.3473** | **0.3639** |

Table 13: Code search results of various architectures and training dataset.

triple quotes ("') or a star slash (\*)). Depending on developer comment habit or docstring style format, docstrings can form two types: one-line docstrings and multi-line (or block) docstrings. A docstring can provide a concise summary of the functionality while also providing a detailed description of the code block, including its parameters, return values, exceptions, and other relevant information (as illustrated in Figure 8)

The primary purpose of a docstring is to provide clear, concise, and easily accessible documentation for a code block. Docstring styles are conventions followed while writing docstrings to ensure consistency, readability, and ease of understanding throughout a codebase. This has become a standard for clean code in the industry and has developers saving tons of time when it comes to understanding or (auto-)generating documentation (using Sphinx, Doxygen, etc).

There are several popular docstring styles, such as Google Style, NumPy Style, reStructuredText (reST) Style for Python programmers, JavaDoc Style or Doxygen for Java users, each with its own formatting rules, structure and target programming language (docstring style examples and preferred language are listed in Figure 10). The statistic for docstring style corresponding to function level is presented in Figure 9. We believe that information inside a docstring is extremely useful and can provide numerous advantages for various applications in the fields of AI for source code, such as providing more precise and relevant search results for code search and retrieval tasks, or the performance of code analysis or refactoring can be significantly improved while the identifier of a parameter and

its corresponding docstring information is available. Furthermore, the presence of various data types allows for the exploration of scenarios such as continual learning [Van et al., 2022, Nguyen et al., 2023, Yadav et al., 2023] and multitask learning [Zhang et al., 2023], which have been lacking investigation in the context of source code data.

## A.6 Experimental results on code summarization

We report Rouge-L, BERTScore, and BLEU-4 metrics on test sets of CSN and The Vault in Table 14. The results obtained from the experiments clearly indicate that models trained on our dataset consistently outperform CSN on all three evaluation metrics. This notable improvement across the metrics serves as strong evidence for the syntactic and semantic richness embedded within our dataset for code summarization. This highlights the effectiveness of our dataset in enabling models to grasp contextual information and generate high-quality summaries that accurately represent the underlying code functionality.

## A.7 Experimental results on code search

In this section, we assess TheVault's versatility and adaptability by providing additional experimental results on several architectures (RoBERTa [Liu et al., 1907], UniXcoder [Guo et al., 2022], PLBART [Ahmad et al., 2021a]) for code search. Tables 13 illustrates the results for code search. As a result, models trained on The Vault consistently outperform all baseline models, underscoring both the efficiency of our pipeline and the dataset's ability to generalize across different architectures.

| Language | Finetune dataset | CodeSearchNet | | | The Vault | | |
|---|---|---|---|---|---|---|---|
| | | Rouge-L | BERTScore | BLEU-4 | Rouge-L | BERTScore | BLEU-4 |
| Python | CodeSearchNet | 34.000 | 88.827 | 19.55 | 26.798 | 87.055 | 10.86 |
| | TheVault/medium-S | 34.676 | 88.905 | 19.74 | 30.335 | 87.633 | 13.06 |
| | TheVault-S | **36.499** | **89.211** | **21.15** | 31.786 | 87.929 | 14.14 |
| | TheVault/medium-L | 33.848 | 88.734 | 18.88 | 30.947 | 87.716 | 13.36 |
| | TheVault-L | 35.024 | 88.921 | 19.83 | **32.251** | **87.954** | **14.33** |
| Java | CodeSearchNet | **35.625** | **89.132** | 20.38 | 27.297 | 87.385 | 8.00 |
| | TheVault/medium-S | 33.385 | 88.490 | 18.62 | 31.320 | 87.897 | 11.17 |
| | TheVault-S | 35.495 | 88.907 | **20.43** | 33.137 | 88.268 | 12.00 |
| | TheVault/medium-L | 32.561 | 88.161 | 18.29 | 30.773 | 87.596 | 11.50 |
| | TheVault-L | 35.221 | 88.782 | 20.37 | 32.882 | 88.000 | **12.47** |
| JavaScript | CodeSearchNet | 28.330 | **87.568** | 16.15 | 24.895 | 86.519 | 8.42 |
| | TheVault/medium-S | 26.528 | 87.017 | 14.88 | 27.891 | 86.846 | 10.58 |
| | TheVault-S | **28.345** | 87.384 | **16.30** | 29.817 | 87.320 | 11.71 |
| | TheVault/medium-L | 27.062 | 87.057 | 14.95 | 28.290 | 86.936 | 10.83 |
| | TheVault-L | 27.869 | 87.276 | 15.63 | **30.572** | **87.391** | **12.38** |
| PHP | CodeSearchNet | **41.346** | **89.981** | **26.26** | 39.960 | 89.281 | 17.85 |
| | TheVault/medium-S | 34.802 | 88.125 | 21.78 | 63.984 | 93.287 | 37.72 |
| | TheVault-S | 37.297 | 88.676 | 23.53 | 65.401 | 93.580 | 38.30 |
| | TheVault/medium-L | 33.325 | 87.963 | 20.27 | 65.195 | 93.679 | 39.13 |
| | TheVault-L | 36.478 | 88.641 | 23.21 | **67.089** | **94.012** | **40.13** |
| Go | CodeSearchNet | 40.076 | 90.487 | 19.83 | 38.189 | 89.994 | 17.87 |
| | TheVault/medium-S | 42.011 | 90.816 | 21.38 | 54.030 | 92.372 | 34.47 |
| | TheVault-S | **44.649** | **91.188** | **24.37** | 54.889 | 92.541 | 35.44 |
| | TheVault/medium-L | 41.480 | 90.731 | 21.22 | 56.721 | 92.994 | 39.27 |
| | TheVault-L | 44.063 | 91.108 | 23.96 | **57.681** | **93.130** | **40.38** |
| Ruby | CodeSearchNet | 28.196 | 87.371 | 15.38 | 24.500 | 86.417 | 10.26 |
| | TheVault/medium-S | 29.680 | 87.559 | 16.09 | 26.904 | 86.964 | 12.26 |
| | TheVault-S | **31.133** | **87.830** | **17.15** | 28.535 | **87.280** | 13.79 |
| | TheVault/medium-L | 29.389 | 87.565 | 15.42 | 27.485 | 87.044 | 12.63 |
| | TheVault-L | 30.634 | 87.759 | 16.53 | **29.141** | 87.223 | **14.24** |
| Total | CodeSearchNet | 36.739 | **89.341** | 21.24 | 30.563 | 87.853 | 16.11 |
| | TheVault/medium-S | 34.935 | 88.755 | 19.91 | 39.589 | 89.278 | 26.02 |
| | TheVault-S | **37.120** | 89.163 | **21.73** | 41.079 | 89.591 | 27.41 |
| | TheVault/medium-L | 34.086 | 88.585 | 19.16 | 40.544 | 89.473 | 27.71 |
| | TheVault-L | 36.305 | 89.024 | 21.14 | **42.187** | **89.753** | **29.32** |
| C | TheVault/medium-S | - | - | - | 28.132 | 86.277 | 10.21 |
| | TheVault-S | - | - | - | 33.275 | 87.353 | 13.39 |
| | TheVault/medium-L | - | - | - | 29.151 | 86.566 | 11.32 |
| | TheVault-L | - | - | - | **35.009** | **87.807** | **14.86** |
| C# | TheVault/medium-S | - | - | - | 39.480 | 89.616 | 23.88 |
| | TheVault-S | - | - | - | **46.854** | **90.819** | **31.11** |
| | TheVault/medium-L | - | - | - | 39.720 | 89.652 | 24.30 |
| | TheVault-L | - | - | - | 46.594 | 90.788 | 31.05 |
| C++ | TheVault/medium-S | - | - | - | 28.029 | 86.719 | 14.55 |
| | TheVault-S | - | - | - | 29.942 | 87.116 | 16.18 |
| | TheVault/medium-L | - | - | - | 28.815 | 86.827 | 14.85 |
| | TheVault-L | - | - | - | **30.754** | **87.163** | **16.65** |
| Rust | TheVault/medium-S | - | - | - | 30.416 | 87.758 | 13.30 |
| | TheVault-S | - | - | - | 32.535 | 88.126 | 14.72 |
| | TheVault/medium-L | - | - | - | 30.999 | 87.862 | 13.75 |
| | TheVault-L | - | - | - | **32.857** | **88.142** | **15.18** |

Table 14: Experimental results for code summarization. For models that are finetuned on The Vault, "-S" annotation refers to finetuning process using *short_docstring* field as summarization, while "-L" represents the *docstring* field.

| Languages | Inconsistent pairs |
|---|---|
| Python | ```python
// Handy for templates.
def has_urls(self):
    if self.isbn_uk or self.isbn_us or self.official_url or self.
     notes_url:
     return True
    else:
     return False
``` |
| | ```python
// compresses the waveform horizontally; one of
// ```"normal"```, ```"resync"```, ```"resync2"```
def phase_type(self, value):
    self._params.phase_type = value
    self._overwrite_lock.disable()
``` |
| Go | ```go
// InWithTags, OutWithTags, Both, BothWithTags
func Predicates(from Shape, in bool) Shape {
    dir := quad.Subject
    if in {
        dir = quad.Object
    }
    return Unique{NodesFrom{
        Quads: Quads{
            {Dir: dir, Values: from},
        },
        Dir: quad.Predicate,
    }}
}
``` |
| | ```go
// select Surf ro PhomtomJS
func (self *DefaultRequest) GetDownloaderID() int {
    self.once.Do(self.prepare)
    return self.DownloaderID
}
``` |
| Java | ```java
// supplied callback function.
public boolean rm(Pipe pipe, IMtrieHandler func, XPub pub)
    {
        assert (pipe != null);
        assert (func != null);
        return rmHelper(pipe, new byte[0], 0, 0, func, pub);
    }
``` |
| | ```java
// only for change appenders
public MapContentType getMapContentType(ContainerType
    containerType){
        JaversType keyType = getJaversType(Integer.class);
        JaversType valueType = getJaversType(containerType.
            getItemType());
        return new MapContentType(keyType, valueType);
    }
``` |

| Languages | Inconsistent pairs |
|-----------|-------------------|
| JavaScript | ```javascript
// we do not need Buffer pollyfill for now
function(str){
 var ret = new Array(str.length), len = str.length;
 while(len--) ret[len] = str.charCodeAt(len);
 return Uint8Array.from(ret);
}
```

```javascript
// WeakMap works in IE11, node 0.12
function (fn, name) {
 function proxiedFn() {
  'use strict';
  var fields = privates.get(this); // jshint ignore:line
  return fn.apply(fields, arguments);
 }

 Object.defineProperty(proxiedFn, 'name', {
  value: name,
  configurable: true
 });

 return proxiedFn;
}
``` |
| PHP | ```php
// -> NEW
public function consumerId()
    {
        if (isset($this->session->data['customer_id']) === true) {
            return $this->session->data['customer_id'];
        }
        return null;
    }
```

```php
// disini mo ba atur akan apa mo kamana
private function _parse_routes()
    {
        $uri=implode('/', $this->uri->segments());

        if (isset($this->router[$uri])) {
            return $this->_set_request(explode('/', $this->router
                [$uri]));
        }

        foreach ($this->router as $key → $val) {
            $key = str_replace(':any', '.+', str_replace(':num',
                '[0-9]+', $key));

            if (preg_match('#^'.$key.'$#', $uri)) {
                if (strpos($val, '$') !== FALSE AND strpos($key,
                    '(') !== FALSE) {
                    $val = preg_replace('#^'.$key.'$#', $val,
                        $uri);
                }

                return $this->_set_request(explode('/', $val));
            }
        }

        $this->_set_request($this->uri->segments());
    }
``` |

| Languages | Inconsistent pairs |
| --- | --- |
| Ruby | ```ruby
// Initialize a new page, which can be simply rendered or
// persisted to the filesystem.
def method_missing(name, *args, &block)
    return meta[name.to_s] if meta.key?(name.to_s)
    super
  end
```

```ruby
// Accepts the path of the YAML file to be parsed into
// commands - will throw a CommandException should it have
// invalid parameters
// @param filePath [String] Path for YAML file
def action_options
    # Attempt resolution to outputs of monitor
    return @action_options unless @monitor_class.outputs.length >
        0
    action_options = @action_options.clone
    @monitor_class.outputs.each do |output, _type|
      action_options.each do |option_key, option_value|
        action_options[option_key] =
          option_value.gsub("{{#{output}}}", @monitor.send(output).
              to_s)
      end
    end
    action_options
  end
``` |

Table 15: Inconsistent pairs in CodeSearchNet found by our model. "//" represents for docstring section.

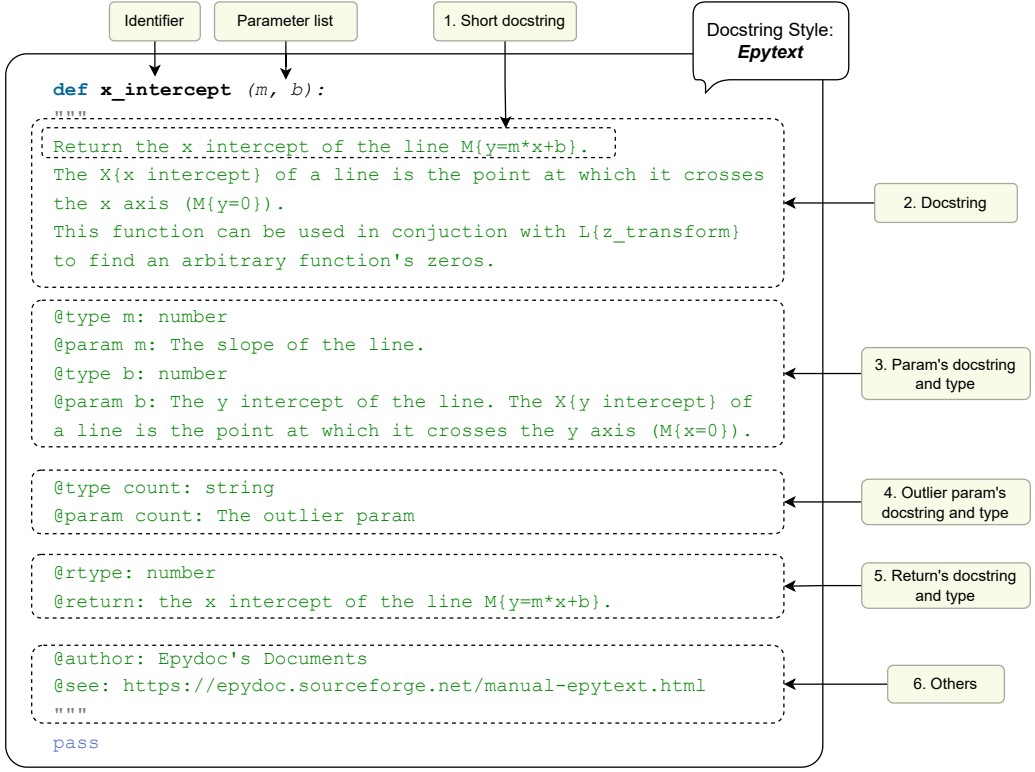

Figure 8: Structure of a docstring and its metadata.

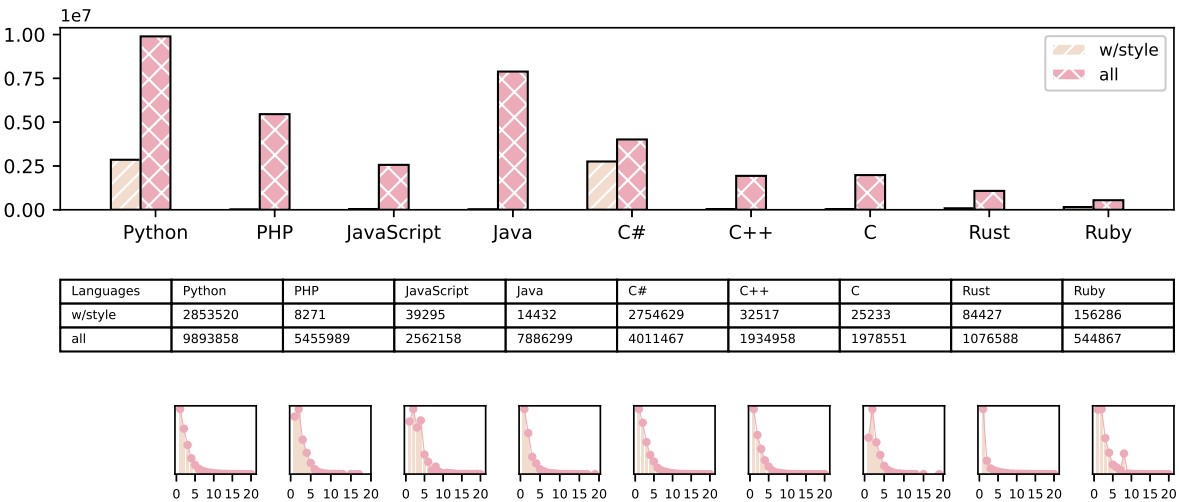

| Languages | Python | PHP | JavaScript | Java | C# | C++ | C | Rust | Ruby |
|-----------|--------|-----|-----------|------|-----|-----|-----|------|------|
| w/style | 2853520 | 8271 | 39295 | 14432 | 2754629 | 32517 | 25233 | 84427 | 156286 |
| all | 9893858 | 5455989 | 2562158 | 7886299 | 4011467 | 1934958 | 1978551 | 1076588 | 544867 |

Figure 9: Number of docstrings follows a specific style over all extracted code-text pairs. **Upper** figure and **Middle** table illustrate statistics for docstrings with style. **Lower** figures present the histogram of extracted attributes in the range of 1-20 for docstrings in each language. Golang does not have a supported style.

## Google Style

Python

```
"""

Test function.
Args:
    param1 (int): Description of param1.
    param2 (str): Description of param2.
Returns:
    bool: Description of the return value.
"""
```

## JavaDoc

Java C C++ C#

```
/**
 * Test function.
 *
 * @param param1 Description of param1.
 * @param param2 Description of param2.
 * @return Description of the return value.
 */
```

## RustDoc

Rust

```
/**
 * Test function.
 ** # Arguments
 * `param1`: Description of param1.
 * `param2`: Description of param2.
 * # Returns
 * Description of the return value.
 */
```

## reST

Python

```
"""Test function.
:param param1: Description of param1.
:type param1: int
:param param2: Description of param2.
:type param2: str
:return: Description of the return value.
:rtype: bool
"""
```

## Rdoc

Ruby

```
=begin
Test method.

@param param1 [Integer] Description of param1.
@param param2 [String] Description of param2.
@return [Boolean] Description of the return value.
=end
```

## Jsdoc

JavaScript

```
/**
 * Test function.
 *
 * @param {int} param1 - Description of param1.
 * @param {string} param2 - Description of param2.
 * @return {bool} Description of the return value.
 */
```

## PHPdoc

PHP

```
/**
 * Test function.
 *
 * @param int $param1 Description of param1.
 * @param string $param2 Description of param2.
 * @return bool Description of the return value.
 */
```

## Doxygen

C C++ C#

```
/**
 * Test function.
 * @brief Constructor.
 * @param param1 Description of param1
 * @param param2 Description of param2
 * @see Test()
 */
```

## XML

C#

```
/// <summary>
/// Test function.
/// </summary>
/// <param name="param1">Description of param1.
</param>
/// <param name="param2">Description of param1.
</param>
/// <returns>
/// Description of the return value.
/// </returns>
```

## Epytext

Python

```
"""
Test function.
@type param1: int
@param param1: Description of param1
@type param2: string
@param param2: Description of param2
@rtype: bool
@return: Description of the return value.
"""
```

## NumPy Style

Python

```
"""
Test function.
Parameters
----------
param1 : int
Description of param1.
param2 : str
Description of param2.
Returns
-------
bool
Description of the return value.
"""
```

Figure 10: Supported docstring styles.