# OpenReview forum: "The Vault: A Comprehensive Multilingual Dataset for Advancing Code Understanding and Generation"
_EMNLP/2023/Conference — EMNLP 2023 Findings_

### Official Review · Reviewer_EqfE · 2023-08-02

**Typos Grammar Style And Presentation Improvements:** 1) Replace 'they' with 'researchers' …
**Soundness:** 3

**Excitement:**

3: Ambivalent: It has merits (e.g., it reports state-of-the-art results, the idea is nice), but there are key weaknesses (e.g., it describes incremental work), and it can significantly benefit from another round of revision. However, I won't object to accepting it if my co-reviewers champion it.

**Missing References:**

The references of BLEU, Rouge-L, and Bertscore metrics.

**Paper Topic And Main Contributions:**

This paper presents a new multilingual code-related dataset named The Vault for training LLMs to understand and generate code. The authors build The Vault on another huge dataset called The Stack and present a novel data cleaning pipeline to process the raw dataset, including the rule-based and dl-based methods. The authors compare the Vault dataset with the baseline CodeSearchNet dataset using various models (CodeBERT, CodeT5, and CodeGen) in different code-related tasks, including code search, summarization, and generation, which finally show improvement.

In conclusion, the main contributions of this paper include:
1) a huge multilingual code dataset
2) a novel data cleaning pipeline improving the quality of code-related datasets

**Questions For The Authors:**

See Reasons To Reject.

**Reasons To Accept:**

1) This paper presents a new pre-process pipeline that could improve code-related datasets' quality.
2) The authors evaluate the proposed Vault dataset across different tasks and models, which shows much improvement compared with the CodeSearchNet dataset.

**Reasons To Reject:**

1) The Vault is built on another already existing Stack dataset, reducing the contribution.
2) The proposed pipeline lacks enough novelty, not only for rules and the trained classifier.
3) The paper needs more details about the D_unimodal and D_block datasets. For example, does the proposed pipeline process these two datasets? Besides, the authors do not evaluate the quality of these two datasets.
4) Some flaws in experiments:\
4.1) The authors compare the Vault and CSN datasets in the code summarization task. However, the authors do not control the sizes of these datasets to be equal or similar. Specifically, the full and medium versions of Vault have ~32 and ~6 million samples, which are much larger than the ~2 million of the CSN (shown in Table 3). As a result, the evaluating results can not firmly prove that the dataset quality of Vault is better than CSN. \
4.2) The paper needs to include the ablation studies for the proposed process pipeline. The following experiments should be considered: the experiment of the un-processed raw Stack dataset vs. Vault; the experiment about the influence of rule-based and dl-based process methods, individually. Likewise, the authors should set the above compared datasets with equal sizes.\
4.3) Besides, the authors could also apply the proposed pipeline to another dataset, such as CSN, and then compare it with the raw dataset, which is helpful to show the generalization ability of these methods.

**Reproducibility:**

4: Could mostly reproduce the results, but there may be some variation because of sample variance or minor variations in their interpretation of the protocol or method.

**Reviewer Confidence:**

4: Quite sure. I tried to check the important points carefully. It's unlikely, though conceivable, that I missed something that should affect my ratings.

---

> ### Author Rebuttal · Authors · 2023-08-29
>
> We genuinely value your thoughtful feedback and constructive questions.
>
> **Q1. The Vault is built on another already existing Stack dataset, reducing the contribution**
>
> The Vault's contribution is in the pipeline of cleaning up and providing high-quality pairs of code and natural language. This pipeline works with any of the large-scale datasets available, including The Pile, The Stack, and CodeParrot.
>
> It should be noted that larger-scale code datasets, such as The Stack, CodeParrot, and The Pile, are collected from public code on Github; the difference between them is mainly in size and some efforts to address privacy concerns, not in data quality. In our case, we could collect public GitHub and use the pipeline to extract high-quality pairs of code and docstring, but this is unnecessary and time-consuming given the availability and comprehensiveness of datasets like The Stack.
>
> **Q2. The proposed pipeline lacks enough novelty, not only for rules and the trained classifier**
>
> Our contributions lie in the rich and diverse information extraction and innovative filtering methods we've introduced. We've implemented a multi-level approach, encompassing functions, inlines, and classes, which effectively addresses the granularity of data extraction. This multilevel type of data provides a valuable foundation for subsequent tasks. A novel assessment approach with carefully grouping examples to prove the effectiveness of our filtering model.
>
> In comparison to existing datasets, we provide a larger set of rules to filter data in reducing noise. In addition, a distinguishing aspect of our work is the application of deep learning techniques to the filtering process. Unlike the mentioned datasets that solely rely on rule-based filtering, we introduce a strategic integration of DL within our pipeline.
>
> **Q3. The paper needs more details about the D_unimodal and D_block datasets. For example, does the proposed pipeline process these two datasets? Besides, the authors do not evaluate the quality of these two datasets**
>
> We acknowledge the reviewer’s observation about the missing experiment in the D_unimodal and D_block datasets. Our main reason for focusing primarily on the D_paired dataset lies in our objective to showcase the dataset's quality and its applicability to various code-related tasks.
>
> To our knowledge, we notice that these two datasets present unique challenges in terms of proper task modeling. Unlike function stand-alone pairs (in D_paired), which are able to demonstrate the benefit of our dataset quality, D_unimodal overlaps with the scope of other raw code collections and often serves as a resource for pre-trained language models [1, 2]. On the other hand, D_block is a lack of clear task definitions or specific problem formulations (e.g., the author in [3] claims class code snippet level might not be as effective for code summarization tasks due to their extended length).
>
> While comprehensive insights and evaluations of these two datasets are not our main focus in this paper, we acknowledge the potential value of future work to delve into the unique characteristics and applications of these datasets.
>
>
> **Q4. Some flaws in experiments:**
>
> **Q4.1. The authors compare the Vault and CSN datasets in the code summarization task. However, the authors do not control the sizes of these datasets to be equal or similar. Specifically, the full and medium versions of Vault have ~32 and ~6 million samples, which are much larger than the ~2 million of the CSN (shown in Table 3). As a result, the evaluating results can not firmly prove that the dataset quality of Vault is better than CSN**
>
> In our Code Search and Code Generation reports, we emphasize the evaluation of TheVault/small; a dataset of comparable size (~1.7M instances) to the CodeSearchNet training set, which presents better performance compared to CSN. It is accurate to acknowledge that we inadvertently omitted experiments with equal dataset sizes in the Code Summarization report. To rectify this, we have included the updated results of our fine-tuning on TheVault-small in the tables below.
>
> - Evaluate on TheVault
> | Model | Fine-tune data | Python | Java | JavaScript | Go | PHP | Ruby | Total/Avg |
> | :--- | :--- | ---: | ---: | ---: | ---: | ---: | ---: | ---: |
> | CodeT5 | CodeSearchNet | 10.86 | 8.00 | 8.42 | 17.87 | 17.85 | 10.26 | 16.11/12.21 |
> |  | TheVault/small | 12.26 | 11.13 | 9.68 | 31.64 | 38.86 | 11.23 | **25.12/19.13** |
> | PLBart | CodeSearchNet | 10.24 | 7.26 | 7.64 | 16.90 | 13.83| 9.60 | 14.39/10.91 |
> |  | TheVault/small | 10.23 | 9.28 | 8.95 | 22.78 | 34.32| 9.74 | **20.29/15.88** |
>
> - Evaluate on CodeSearchNet (BLEU-4)
> | Model | Fine-tune data | Python | Java | JavaScript | Go | PHP | Ruby | Total/Avg |
> | :--- | :--- | ---: | ---: | ---: | ---: | ---: | ---: | ---: |
> | CodeT5 | CodeSearchNet | 19.55 | 20.38 | 16.15 | 19.83 | 26.26 | 15.38 | **21.24/19.59** |
> |  | TheVault/small | 18.94 | 17.72 | 13.96 | 19.92 | 20.43 | 15.22 | 18.83/17.70 |
> | PLBart | CodeSearchNet | 17.99 | 17.38| 14.84 | 17.98 | 22.54 | 14.08 | **18.78/17.47** |
> |  | TheVault/small | 14.93 | 15.66 | 11.95 | 17.03 | 18.0 | 11.49 | 15.95/14.84 |
>
> Even though we observe the result evaluation on CodeSearchNet testset is lower, the model trained on TheVault outperform with a more significant gap in TheVault testset. This phenomenon can be explained by the mention of the post-processing step to reduce the difference of the CSN and TheVault filtering methods (lines 418-421) in our paper, where the syntactic distribution can still exhibit nonidentical characteristics, which can affect the BLEU score. However, this gap can be overcome by using the full dataset of The Vault to achieve better results in both test sets.
>
>
> **Q4.2. The paper needs to include the ablation studies for the proposed process pipeline. The following experiments should be considered: the experiment of the un-processed raw Stack dataset vs. Vault; the experiment about the influence of rule-based and dl-based process methods, individually. Likewise, the authors should set the above compared datasets with equal sizes**
>
> In the experiment of code generation, we had already compared the unprocessed The Stack (Multi-TheVault-raw) dataset and The Vault  (Table 7). We provide additional results for code summarization and code search tasks in the below tables. As a result, the performance of models trained on The Vault surpasses trained models on The Stack, presenting the effectiveness of our cleaning process.
>
> In order to address your concern, we randomly select subsets from the raw code-text function pairs that were retrieved from The Stack, without performing any filtering (raw-TheStack), each set approximately the size of TheVault/small (~1.7M code-text instances). We use 3 different seeds to sample raw-TheStack and report the average result. The impact of our cleaning procedure on raw, extracted The Stack is presented in the tables below; these comparisons show that models trained on The Vault significantly outperform models that are trained on raw-TheStack.
>
> *Code Search*
> - Evaluation on TheVault (MRR)
> | Model | Python | Java | JavaScript | Go | PHP | Ruby | Rust | C++ | C | C# |
> | --- | --- | --- | --- | --- | --- | --- | --- | --- | --- | --- |
> | raw-TheStack | 0.318 | 0.3245 | 0.1837 | 0.4194 | 0.1718 | 0.0878 | 0.1949 | 0.2024 | 0.2204 | 0.2309 |
> | theVault-small | **0.3501** | **0.4214** | **0.3216** | **0.4864** | **0.2351** | **0.2904** | **0.326** | **0.2996** | **0.3015** | **0.3483** |
>
> - Evaluation on CodeSearchNet (MRR)
> | Model | Python | Java | JavaScript | Go | PHP | Ruby |
> | --- | --- | --- | --- | --- | --- | --- |
> | raw-TheStack | 0.3713 | 0.3492 | 0.3148 | 0.5519 | 0.2731 | 0.2748 |
> | theVault-small | **0.4074** | **0.4857** | **0.4466** | **0.6578** | **0.6578** | **0.5251** |
>
> *Code Summarization*
> - Evaluate on The Vault (BLEU-4)
> | Model | Fine-tune data | Python | Java | JavaScript | Go | PHP | Ruby | Total/Avg |
> | :--- | :--- | ---: | ---: | ---: | ---: | ---: | ---: | ---: |
> | CodeT5 | raw-TheStack | 16.18 | 9.06 | 6.23| 19.05 | 7.07 | 5.78 | 11.84/10.56 |
> |  | TheVault/small |12.26 | 11.13 | 9.68 | 31.64 | 38.86 | 11.23 | **25.12/19.13** |
> | PLBart | raw-TheStack | 1.69  | 4.02 | 0.43 | 24.60  | 4.83  | 0.49  | 07.19/06.01 |
> |  | TheVault/small | 10.23 | 9.28 | 8.95 | 22.78 | 34.32| 9.74 | **20.29/15.88** |
>
> - Evaluate on CodeSearchNet (BLEU-4)
> | Model | Fine-tune data | Python | Java | JavaScript | Go | PHP | Ruby | Total/Avg |
> | :--- | :--- | ---: | ---: | ---: | ---: | ---: | ---: | ---: |
> | CodeT5 | raw-TheStack | 16.18 | 9.06 | 6.23| 19.05 | 7.07 | 5.78 | 11.84/10.56 |
> |  | TheVault/small | 18.94 | 17.72 | 13.96 | 19.92 | 20.43 | 15.22 | **18.83/17.70**|
> | PLBart | raw-TheStack | 0.86 | 3.06 | 0.59 | 10.91 | 2.29 | 0.47 | 3.23/3.03 |
> |  | TheVault/small | 14.93 | 15.66 | 11.95 | 17.03 | 18.0 | 11.49 | **15.95/14.84** |
>
> Furthermore, the practical instances showcased in Table 8 and Table 14 within our paper provide a rational basis for both our rule-based and deep learning-based methodologies, respectively.
>
> **Q4.3. Besides, the authors could also apply the proposed pipeline to another dataset, such as CSN, and then compare it with the raw dataset, which is helpful to show the generalization ability of these methods.**
>
> We acknowledge your recommendation to apply the pipeline to CSN. However, there are several difficulties since CSN is already processed and filtered from raw GitHub files in different initial steps. Thus, we can not filter CSN from beginning. Table 8 and Table 14 within our paper apply to CSN can show a rational basis of our filtering approach.
>
> We refer that the comparison experiment to apply the pipeline to CSN could be simulated by comparing the raw version of The Stack (raw-TheStack) and TheVault/small, since they have similar in size and are sourced from GitHub. The results are shown in Q4.2 indicate the effectiveness of our process.
>
> [1] Nijkamp, E., Pang, B., Hayashi, H., Tu, L., Wang, H., Zhou, Y., ... & Xiong, C. (2022, September). CodeGen: An Open Large Language Model for Code with Multi-Turn Program Synthesis. In The Eleventh International Conference on Learning Representations. \
> [2] Wang, Y., Le, H., Gotmare, A. D., Bui, N. D., Li, J., & Hoi, S. C. (2023). Codet5+: Open code large language models for code understanding and generation. arXiv preprint arXiv:2305.07922. \
> [3] Parvez, M. R., Ahmad, W. U., Chakraborty, S., Ray, B., & Chang, K. W. (2021). Retrieval augmented code generation and summarization. arXiv preprint arXiv:2108.11601. \

---

### Official Review · Reviewer_rHmi · 2023-08-03

**Paper Topic And Main Contributions:** 1. This paper presents a new dataset …
**Typos Grammar Style And Presentation Improvements:** 1. In line 245, The ')' character has…
**Soundness:** 3

**Excitement:**

2: Mediocre: This paper makes marginal contributions (vs non-contemporaneous work), so I would rather not see it in the conference.

**Missing References:**

N/A

**Questions For The Authors:**

A. This paper mentions obtaining a consistency score through analogous prediction with GPT-3.5. It would be better if this paper could provide a more detailed explanation of how the authors perform the analogous prediction. Showing a prompting template or a prompt example would be very helpful.

B. In Table 1, various rule-based filters are categorized into three categories: Readability, Consistency, and Intended usage. However, what practical significance does this tripartite categorization have? And the criteria for classification do not seem clear. For example, in the case of Strip Hyperlink, this paper classifies it as promoting consistency, but it seems that it could also be categorized as enhancing readability.

C. Why is only one base model used for each task? In this paper, the CodeBERT model is used for the code search task, the CodeT5 model for the summarization task, and the Codegen multi-350m model for the code generation task. It would be better to use a few baselines for each task instead of only one.

D. Does the deep-learning based filter model produce an output in the form of a two-dimensional vector?

E. In line 890, this paper mentions that the lowest top 50 from each class are chosen when selecting the uncertainty group. But if there exists a data sample where the score for class 0 is very low but the score for class 1 is very high, wouldn't that sample be included in both the Consistency group and the Uncertainty group?


**Reasons To Accept:**

1. The issues with the data quality and quantity of the existing code datasets are well analyzed, and practical solutions are proposed well.

2. The performance improvement is noticeable. When performing the code summarization task and code search task, a model trained on the training set of Vault performs better on the testing set of Code Search Net (CSN), than a model trained on the training set of CSN.


**Reasons To Reject:**

1. For the code summarization task and the code search task, the dataset size of Vault and CSN should be the same in order to demonstrate that Vault is better in terms of data quality than CSN.

2. The compared dataset, CSN, is relatively old. It would be more convincing if the authors could include more recent datasets such as XLCoST for comparison.

3. The authors run experiments on three tasks only: Code Search, Code Summarization, and Code Generation. To show the general performance improvement, it would require to include more tasks such as code translation or completion.

4. In the experiments, only one baseline model is considered for each task. It would be better if there are more baselines to show the generality of Valut. Without it, one might suspect that only the well-performing results are reported.


**Reproducibility:**

3: Could reproduce the results with some difficulty. The settings of parameters are underspecified or subjectively determined; the training/evaluation data are not widely available.

**Reviewer Confidence:**

4: Quite sure. I tried to check the important points carefully. It's unlikely, though conceivable, that I missed something that should affect my ratings.

---

> ### Author Rebuttal · Authors · 2023-08-29
>
> We extend our sincere gratitude your valuable insights and thoughtful evaluation of our work.
>
> **Q1. For the code summarization task and the code search task, the dataset size of Vault and CSN should be the same in order to demonstrate that Vault is better in terms of data quality than CSN.**
>
> We understand your concern about the uneven and unfairness between experimenting dataset. In table 12, we highlight that TheVault/small (~1.7M instances) which is comparable in size to CodeSearchNet training set and in the Table 6 and 7, we reported that model trained on TheVault/small is better than model trained on CodeSearchNet in task code search and code generation.
>
> For code summarization, we update the result in the table below. The model trained on TheVault outperform those trained on CodeSearchNet with a significant gap in TheVault testset, however, we expect the evaluation result on CodeSearchNet’s testset will be lower. Explain to this is our mentioned of the post-processing step to reduce the difference of the CSN and TheVault filtering methods (lines 418-421 in our paper), where the syntactic distribution can still exhibit nonidentical characteristics, which can affect the BLEU score.
>
> *Code Summarization*
> * Evaluate on TheVault
> | Model | Fine tune data | Python | Java | JavaScript | Go | PHP | Ruby | Total/Avg |
> | :--- | :--- | ---: | ---: | ---: | ---: | ---: | ---: | ---: |
> | CodeT5 | CSN | 10.86 | 8.00 | 8.42 | 17.87 | 17.85 | 10.26 | 16.11/12.21 |
> |  | TheVault/small | 12.26 | 11.13 | 9.68 | 31.64 | 38.86 | 11.23 | **25.12/19.13** |
>
>
> * Evaluate on CSN
> | Model | Fine tune data | Python | Java | JavaScript | Go | PHP | Ruby | Total/Avg |
> | :--- | :--- | ---: | ---: | ---: | ---: | ---: | ---: | ---: |
> | CodeT5 | CSN | 19.55 | 20.38 | 16.15 | 19.83 | 26.26 | 15.38 | **21.24/19.59** |
> |  | TheVault/small | 18.94 | 17.72 | 13.96 | 19.92 | 20.43 | 15.22 | 18.83/17.70 |
>
> **Q2. The compared dataset, CSN, is relatively old. It would be more convincing if the authors could include more recent datasets such as XLCoST for comparison.**
>
> CodeSearchNet is a well-known and widely recognized dataset in the field, known for its quality and usability, and it covers a wide range of languages. Current state-of-the-art models (e.g., CodeT5+ [1]) continue to benefit from and are evaluated on this dataset from CodeSearchNet. In contrast, the XLCost dataset is relatively new and has not been widely used as a benchmark for Code Large Language Models. Furthermore, no analysis in XLCost shows that the quality of XLCost is higher than CSN, so there are no compelling reasons to benchmark on XLCost. However, we recognize the importance of comparing the most recent datasets, and we will perform additional comparisons with XLCost in our future extension.
>
> **Q3. The authors run experiments on three tasks only: Code Search, Code Summarization, and Code Generation. To show the general performance improvement, it would require to include more tasks such as code translation or completion.**
>
> We want to highlight that The Vault introduces large-scale code-text pairs data, which facilitates a deeper understanding of the relationships between programming languages and natural language. Consequently, we chose to focus our analysis on Code Search, Code Summarization, and Code Generation tasks, as they inherently encapsulate the integration of code and natural language. While tasks like code translation or code completion are undoubtedly important and interesting, they involve code-to-code interactions that fall beyond the primary scope and objectives of The Vault dataset.
>
>
> **Q4. In the experiments, only one baseline model is considered for each task. It would be better if there are more baselines to show the generality of Vault. Without it, one might suspect that only the well-performing results are reported.**
>
> We acknowledge your concern about the comprehensiveness of our assessment due to the inclusion of a single model in each downstream task. While some existing datasets in related works might explore a variety of models, they often lack a thorough evaluation of data quality compared to other datasets. In contrast, we spend the effort on data assessment of The Vault to other datasets, hence, we fixed model architecture.
>
> To address your concern, we provide experiment results on additional several models (RoBERTa [2], UniXcoder [3], PLBART [4]) for code search, code summarization and a large-scale model CodeGen-2B-multi for code generation task. We train on both *short_docstring* and *docstring* and report the best setting. As a result, We witness a similar trend across various architectures.
>
> *Code Search*
> * Evaluation on TheVault (MRR)
> | Model | Fine-tune data | Python | Java | JavaScript | Go | PHP | Ruby | Avg |
> | --- | --- | --- | --- | --- | --- | --- | --- | --- |
> | CodeBERT | CodeSearchNet | 0.2881 | 0.3213 | 0.2409 | 0.4123 | 0.1854 | 0.2579 | 0.2843 |
> |  | TheVault/small | 0.3501 | 0.4214 | 0.3216 | 0.4864 | 0.2351 | 0.2904 | **0.3508** |
> | RoBERTa | CodeSearchNet | 0.2644 | 0.3329 | 0.2371 | 0.2375 | 0.1577 | 0.2574 | 0.2478 |
> |  | TheVault/small | 0.4533 | 0.5519 | 0.4386 | 0.5021 | 0.2876 | 0.3717 | **0.4342** |
> | UniXCoder | CodeSearchNet | 0.2959 | 0.344 | 0.2508 | 0.185 | 0.1646 | 0.2669 | 0.2512 |
> |  | TheVault/small | 0.3852 | 0.4279 | 0.3491 | 0.4628 | 0.238 | 0.3201 | **0.3639** |
>
> * Evaluation on CodeSearchNet (MRR)
> | Model | Fine-tune data | Python | Java | JavaScript | Go | PHP | Ruby | Avg |
> | --- | --- | --- | --- | --- | --- | --- | --- | --- |
> | CodeBERT | CodeSearchNet | 0.3793 | 0.4636 | 0.4437 | 0.6201 | 0.4741 | 0.5219 | 0.4838 |
> |  |  TheVault/small | 0.4074 | 0.4857 | 0.4466 | 0.6578 | 0.6578 | 0.5251 | **0.5301** |
> | RoBERTa | CodeSearchNet | 0.3479 | 0.448 | 0.4254 | 0.5684 | 0.4623 | 0.5147 | 0.4611 |
> |  | TheVault/small | 0.4849 | 0.5581 | 0.4962 | 0.7446 | 0.5166 | 0.59 | **0.5651** |
> | UniXCoder | CodeSearchNet | 0.3935 | 0.4549 | 0.4459 | 0.5861 | 0.489 | 0.5446 | 0.4857 |
> |  | TheVault/small | 0.4427 | 0.4909 | 0.4506 | 0.6416 | 0.4515 | 0.5702 | **0.5079** |
>
>
> *Code Summarization*
> * Evaluate on TheVault (BLEU-4)
> | Model | Fine-tune data | Python | Java | JavaScript | Go | PHP | Ruby | Total/Avg |
> | :--- | :--- | ---: | ---: | ---: | ---: | ---: | ---: | ---: |
> | CodeT5 | CodeSearchNet | 10.86 | 8.00 | 8.42 | 17.87 | 17.85 | 10.26 | 16.11/12.21 |
> |  | TheVault/small | 12.26 | 11.13 | 9.68 | 31.64 | 38.86 | 11.23 | **25.12/19.13** |
> | PLBART | CodeSearchNet | 10.24 | 7.26 | 7.64 | 16.90 | 13.83| 9.60 | 14.39/10.91 |
> |  | TheVault/small | 10.23 | 9.28 | 8.95 | 22.78 | 34.32| 9.74 | **20.29/15.88** |
>
> * Evaluate on CodeSearchNet (BLEU-4)
> | Model | Fine-tune data | Python | Java | JavaScript | Go | PHP | Ruby | Total/Avg |
> | :--- | :--- | ---: | ---: | ---: | ---: | ---: | ---: | ---: |
> | CodeT5 | CodeSearchNet | 19.55 | 20.38 | 16.15 | 19.83 | 26.26 | 15.38 | **21.24/19.59** |
> |  | TheVault/small | 18.94 | 17.72 | 13.96 | 19.92 | 20.43 | 15.22 | 18.83/17.70 |
> | PLBART | CodeSearchNet | 17.99 | 17.38| 14.84 | 17.98 | 22.54 | 14.08 | **18.78/17.47** |
> |  | TheVault/small | 14.93 | 15.66 | 11.95 | 17.03 | 18.0 | 11.49 | 15.95/14.84 |
>
>
> *Code Generation*
> |Model | Model size |  Dataset | pass@1 | pass@10 | pass@100 |
> |:-------|:------|:------|------:|-------:|-------:|
> |CodeGen-multi | 350m | HumanEval |  6.67 | 10.61| 16.84 |
> |Multi-TheVault | 350m | HumanEval |  **8.14** | **18.12** | **30.0** |
> |CodeGen-multi | 2B | HumanEval |  **14.51** | 24.67 | 38.56 |
> |Multi-TheVault | 2B | HumanEval |  14.00 | **25.74** | **41.72** |
> |CodeGen-multi | 350m | MBPP| 7.46| 24.18 | 46.37 |
> |Multi-TheVault | 350m | MBPP |  **10.13** | **33.96** | **53.20** |
> |CodeGen-multi | 2B | MBPP |  18.06 | 45.80 | **65.34** |
> |Multi-TheVault | 2B | MBPP |  **27.82** | **50.06** | 65.06 |
>
> **Q5. This paper mentions obtaining a consistency score through analogous prediction with GPT-3.5. It would be better if this paper could provide a more detailed explanation of how the authors perform the analogous prediction. Showing a prompting template or a prompt example would be very helpful.**
>
> We provide the template that is used to request GPT-3.5-turbo and some output examples as follows. We will also release these samples and GPT output in our public repository.
>
> **Prompt**
> ```python
> I want you to act as an unbiased docstring evaluator for code. I will give you a docstring along with a source code, and you will give me a score for the consistency between them. The score will be on a scale of 1 to 10, 10 means the docstring can effectively summarize the code while 1 means they are inconsistent. The response answers must contain the score and the explanation that follows the format in the response format.
>
> - Response format:
> Score: X
> Explanation: Y
>
> - Docstring:
> "{docstring}"
>
> - Code:
> "{code}"
> ```
> **Examples**
> ```
> - Prediction: Consistence
> - Group: Consistence
> - Docstring: Reads in a GSLIB-style Geo-EAS data file
> - Code:
> def read_gslib(flname, headeronly=False, delimiter=r'\s*', tmin=None):
>     _test_file_open(flname)
>     if headeronly:
>         nrows = 1
>     else:
>         nrows = None
>     with open(flname, 'r') as datafl:
>         _ = datafl.readline().strip()
>         nvar = int(datafl.readline().split()[0])
>         varnames = []
>         for _ in range(nvar):
>             varnames.append(datafl.readline().strip())
>         nrowtoskip = 2 + nvar
>     engine = Parameters['data.io.pandas_engine']
>     try:
>         tpdf = pd.read_csv(flname, skiprows=nrowtoskip, header=None,
>                            delimiter=delimiter, skipinitialspace=True,
>                            nrows=nrows, engine=engine, chunksize=100000)
>         data = pd.concat(tpdf, ignore_index=True)
>     except (ValueError, NotImplementedError):
>         data = pd.read_csv(flname, skiprows=nrowtoskip, header=None, delimiter=r'\s*',
>                            skipinitialspace=True, nrows=nrows, engine='python')
>     data = _data_trim(data, tmin=tmin)
>     data.columns = varnames
>     return data
> - GPT output:
> Score: 8
> Explanation: The docstring states that the function reads in a GSLIB-style Geo-EAS data file, which is consistent with what the code does. However, the docstring could be more detailed and provide information on the expected inputs and outputs of the function.
> ```
>
> ```
> - Prediction: Inconsistence
> - Group: Uncertainty
> - Docstring: Here we make the assumption that all functions are not native
> - Code:
> public Tuple<String, LibraryFunction> call(String library, String[] args) {
>                 String funct_name = "__" + name.toUpperCase() + "__";
>                 String[] used_args = usedArgs(args);
>                 String funct_call = ((args[0] == null) ? "" : Services.strip(args[0]) + " ") +
>                         funct_name + "(" + Services.join(used_args, ", ") + ")" +
>                         ((args[args.length - 1] == null) ? "" : " " + args[args.length - 1]);
>                 PreprocessData.addPlaceholderFunction(funct_name);
>                 return new Tuple<String, LibraryFunction>(funct_call, this);
>         }
>
>
> - GPT output:
> Score: 6
> Explanation: The docstring mentions an assumption about the functions, but it is not clear how this assumption relates to the code. The code itself seems to be a method that takes a library name and an array of arguments, and returns a tuple containing a function call and a LibraryFunction object. There is no clear connection between the docstring and the code, so the score is 6.
> ```
>
> **Q6. In Table 1, various rule-based filters are categorized into three categories: Readability, Consistency, and Intended usage. However, what practical significance does this tripartite categorization have? And the criteria for classification do not seem clear. For example, in the case of Strip Hyperlink, this paper classifies it as promoting consistency, but it seems that it could also be categorized as enhancing readability.**
>
> The three categories correspond with our data cleaning objectives, which we derived through analyzing the dataset and closely monitoring the outcomes yielded by models trained on the dataset. Categorization gave us practical guidelines to brainstorm and implement rule-based filters. We made available on GitHub the source code for our data cleaning framework so that the community can tailor it to build custom filters that meet their own data cleaning objectives.
>
> **Q7. Why is only one base model used for each task? In this paper, the CodeBERT model is used for the code search task, the CodeT5 model for the summarization task, and the Codegen multi-350m model for the code generation task. It would be better to use a few baselines for each task instead of only one.**
>
> We have updated the additional experiments. Please refer to Q4.
>
> **Q8. Does the deep-learning based filter model produce an output in the form of a two-dimensional vector?**
>
> Yes, the output in the form of a two-dimensional vector as it is a binary classification problem. The model is trained to classify if a pair of code-text is aligned or not.
>
> **Q9. In line 890, this paper mentions that the lowest top 50 from each class are chosen when selecting the uncertainty group. But if there exists a data sample where the score for class 0 is very low but the score for class 1 is very high, wouldn't that sample be included in both the Consistency group and the Uncertainty group?**
>
> The Uncertainty group aims to select the samples that introduce high entropy in the model prediction (the output vector is around 0.5 in each class). For each class, we select samples that have the lowest score (which is always higher than 0.5) based on the prediction probability for the class they belong to create the Uncertainty group.
>
> The case where the score for one class is notably low while another class has a significantly high score can fall within the Uncertainty category if the model exhibits high confidence in its predictions even on the examples that have low scores. However, we avoid overlapping between each group by not repeating the selection of examples already selected in another set. The snippet code below shows our selection strategy.
>
> ```python
> unmatch_set = sorted([x for x in data if x[“class”] == 0], key= lambda x: x[“prediction”][1])
> match_set = sorted([x for x in data if x[“class”] == 1], key= lambda x: -x[“prediction”][1])
>
> unmatch_subset = unmatch_set[:num_samples]
> match_set_subset = match_set[:num_samples]
> uncertain_set = unmatch_set[num_samples:][(-num_samples//2):] + match_set[num_samples:][(-num_samples//2):]
> ```
>
>
>
> [1] Wang, Y., Le, H., Gotmare, A. D., Bui, N. D., Li, J., & Hoi, S. C. (2023). Codet5+: Open code large language models for code understanding and generation. arXiv preprint arXiv:2305.07922. \
> [2] Liu, Y., Ott, M., Goyal, N., Du, J., Joshi, M., Chen, D., ... & Stoyanov, V. (2019). Roberta: A robustly optimized bert pretraining approach. arXiv preprint arXiv:1907.11692. \
> [3] Guo, D., Lu, S., Duan, N., Wang, Y., Zhou, M., & Yin, J. (2022). Unixcoder: Unified cross-modal pre-training for code representation. arXiv preprint arXiv:2203.03850. \
> [4] Ahmad, W. U., Chakraborty, S., Ray, B., & Chang, K. W. (2021). Unified pre-training for program understanding and generation. arXiv preprint arXiv:2103.06333.

---

### Official Review · Reviewer_H3Qp · 2023-08-05

**Soundness:** 3

**Excitement:**

3: Ambivalent: It has merits (e.g., it reports state-of-the-art results, the idea is nice), but there are key weaknesses (e.g., it describes incremental work), and it can significantly benefit from another round of revision. However, I won't object to accepting it if my co-reviewers champion it.

**Missing References:**

None

**Paper Topic And Main Contributions:**

The paper introduces "The Vault," a dataset tailored for training large language models to understand and generate code. The dataset consists of 41 million pairs of high-quality code-text pairs extracted from 10 prevalent programming languages.

It surpasses existing datasets both in quality and scale. Empirical evaluation of "The Vault" by finetuning popular code language models, illustrating its superior performance compared to other datasets. And the entire toolkit used for dataset creation and quality assurance open-sourced to foster community engagement.

**Questions For The Authors:**

A. Given that both CodeSearchNet and The Vault derive significant code samples from GitHub, have you assessed any overlap between the test set of one and the training set of the other to prevent potential data leakage?

B. Recognizing that CodeSearchNet and The Stack are distinct datasets, how might model performance in code search and code summarization tasks differ if we were to draw random text-code pairs from The Stack as a baseline, in comparison to your refined The Vault?

C. The term "Multi-TheVault-raw" in Table 7 isn't elaborated upon within the paper. Is this intended to represent a dataset randomly sampled from The Stack?

**Reasons To Accept:**

- The paper introduces a substantially larger dataset of high-quality code-text pairs, surpassing existing datasets like CodeSearchNet. Evidence from three downstream tasks indicates that models trained on the Vault outperform those trained on CodeSearchNet.
- The authors present an innovative data cleaning methodology that blends traditional rule-based techniques with deep learning classifiers, effectively filtering out many inconsistent code-text pairs.
- By making the toolkit for dataset creation and cleaning publicly available, the authors not only show dedication to open research but also encourage reproducibility and further community-driven improvements.
- The paper is well written and easy to follow. Detailed experiments are conducted to help evaluate and analyze the capabilities of the new dataset.

**Reasons To Reject:**

- Although the dataset includes 10 widely-used languages, they are just a fraction of the 300 languages found in The Stack. This could reduce its attractiveness for researchers focused on less prevalent languages.
- The code generation evaluation was based on small models like CodeGen 350M. This could restrict our understanding of the dataset's efficacy with large-scale models.
- Only one model is inluced in each down stream task, which may not provide a comprehensive assessment of the dataset's versatility and adaptability across different architectures and frameworks.
- Even though The Vault emphasizes quality, its scale is considerably smaller than some raw datasets like The Stack and The Pile. This may limit its effectiveness in some applications where sheer volume is paramount.

**Reproducibility:**

5: Could easily reproduce the results.

**Reviewer Confidence:**

5: Positive that my evaluation is correct. I read the paper very carefully and I am very familiar with related work.

**Typos Grammar Style And Presentation Improvements:**

- “CodeBert” in L450 and the title of Table 10 should be “CodeBERT”.
- “Codegen” in L459, L468 and table 7 should be “CodeGen”.

---

> ### Author Rebuttal · Authors · 2023-08-29
>
> First of all, we would like to express our gratitude for your constructive feedback.
>
> **Q1. Although the dataset includes 10 widely-used languages, they are just a fraction of the 300 languages found in The Stack. This could reduce its attractiveness for researchers focused on less prevalent languages.**
>
> We acknowledge your point about our toolkit's limited number of supported languages. There are a few obstacles that prevent us from expanding our dataset beyond the top ten popular languages:
> 1) Because each programming language has its own AST parser and grammar, and tree-sitter is a parser generator tool that provides a platform for users to write grammar for each language and. As a result, not every language in The Stack's list of 300 has an AST parser.
> 2) Our primary effort is on parsing the code into AST for each programming language using a specific parser and extract code-text pairs from the ASTs. Beyond the scope of ten languages, writing the parser for the missing ones (among 300) is extremely time-consuming.
>
> We did investigate the possibility of including low-resource languages in addition to the initial set of ten popular languages. During our testing, however, we encountered difficulties in maintaining control over the quality and consistency of parsing across a wide range of languages. As a result, our strategy is to bundle our toolkit into a library that can be easily extended so that the research community can contribute to the expansion to other languages.
>
> **Q2. The code generation evaluation was based on small models like CodeGen 350M. This could restrict our understanding of the dataset's efficacy with large-scale models.**
>
> We provide the results of CodeGen 2B and the additional MBPP dataset in the code generation experiment. As a result, The Vault can improve the performance of pre-existing pre-trained models in both datasets.
>
> |Model | Model size |  Dataset | pass@1 | pass@10 | pass@100 |
> |:-------|:------|:------|------:|-------:|-------:|
> |CodeGen-multi | 350m | HumanEval |  6.67 | 10.61| 16.84 |
> |Multi-TheVault | 350m | HumanEval |  **8.14** | **18.12** | **30.0** |
> |CodeGen-multi | 2B | HumanEval |  **14.51** | 24.67 | 38.56 |
> |Multi-TheVault | 2B | HumanEval |  14.00 | **25.74** | **41.72** |
> |CodeGen-multi | 350m | MBPP| 7.46| 24.18 | 46.37 |
> |Multi-TheVault | 350m | MBPP |  **10.13** | **33.96** | **53.20** |
> |CodeGen-multi | 2B | MBPP |  18.06 | 45.80 | **65.34** |
> |Multi-TheVault | 2B | MBPP |  **27.82** | **50.06** | 65.06 |
>
>
> **Q3. Only one model is included in each down stream task, which may not provide a comprehensive assessment of the dataset's versatility and adaptability across different architectures and frameworks.**
>
> We acknowledge your concern about the comprehensiveness of our assessment due to the inclusion of a single model in each downstream task. While some existing datasets in related works might explore a variety of models, they often lack a thorough evaluation of data quality compared to other datasets. In contrast, we spend the effort on data assessment of The Vault to other datasets, hence, we fixed model architecture.
>
> To address your concern about TheVault’s versatility and adaptability, we provide experiment results on several architectures (RoBERTa [5], UniXcoder [6], PLBART [7]) for code search and code summarization (and a large-scale model in the code generation task mentioned in the Q2). As a result, We witness a similar trend across various architectures.
>
> *Code Search:*
> - Evaluation on TheVault (MRR)
> | Model | Fine-tune data | Python | Java | JavaScript | Go | PHP | Ruby | Avg |
> | --- | --- | --- | --- | --- | --- | --- | --- | --- |
> | CodeBERT | CodeSearchNet | 0.2881 | 0.3213 | 0.2409 | 0.4123 | 0.1854 | 0.2579 | 0.2843 |
> |  | TheVault/small | 0.3501 | 0.4214 | 0.3216 | 0.4864 | 0.2351 | 0.2904 | **0.3508** |
> | RoBERTa | CodeSearchNet | 0.2644 | 0.3329 | 0.2371 | 0.2375 | 0.1577 | 0.2574 | 0.2478 |
> |  | TheVault/small | 0.4533 | 0.5519 | 0.4386 | 0.5021 | 0.2876 | 0.3717 | **0.4342** |
> | UniXCoder | CodeSearchNet | 0.2959 | 0.344 | 0.2508 | 0.185 | 0.1646 | 0.2669 | 0.2512 |
> |  | TheVault/small | 0.3852 | 0.4279 | 0.3491 | 0.4628 | 0.238 | 0.3201 | **0.3639** |
>
> - Evaluation on CodeSearchNet (MRR)
> | Model | Fine-tune data | Python | Java | JavaScript | Go | PHP | Ruby | Avg |
> | --- | --- | --- | --- | --- | --- | --- | --- | --- |
> | CodeBERT | CodeSearchNet | 0.3793 | 0.4636 | 0.4437 | 0.6201 | 0.4741 | 0.5219 | 0.4838 |
> |  |  TheVault/small | 0.4074 | 0.4857 | 0.4466 | 0.6578 | 0.6578 | 0.5251 | **0.5301** |
> | RoBERTa | CodeSearchNet | 0.3479 | 0.448 | 0.4254 | 0.5684 | 0.4623 | 0.5147 | 0.4611 |
> |  | TheVault/small | 0.4849 | 0.5581 | 0.4962 | 0.7446 | 0.5166 | 0.59 | **0.5651** |
> | UniXCoder | CodeSearchNet | 0.3935 | 0.4549 | 0.4459 | 0.5861 | 0.489 | 0.5446 | 0.4857 |
> |  | TheVault/small | 0.4427 | 0.4909 | 0.4506 | 0.6416 | 0.4515 | 0.5702 | **0.5079** |
>
> *Code Summarization:*
> - Evaluate on TheVault (BLEU-4)
> | Model | Fine-tune data | Python | Java | JavaScript | Go | PHP | Ruby | Total/Avg |
> | :--- | :--- | ---: | ---: | ---: | ---: | ---: | ---: | ---: |
> | CodeT5 | CodeSearchNet | 10.86 | 8.00 | 8.42 | 17.87 | 17.85 | 10.26 | 16.11/12.21 |
> |  | TheVault/small | 12.26 | 11.13 | 9.68 | 31.64 | 38.86 | 11.23 | **25.12/19.13** |
> | PLBART | CodeSearchNet | 10.24 | 7.26 | 7.64 | 16.90 | 13.83| 9.60 | 14.39/10.91 |
> |  | TheVault/small | 10.23 | 9.28 | 8.95 | 22.78 | 34.32| 9.74 | **20.29/15.88** |
>
> - Evaluate on CodeSearchNet (BLEU-4)
> | Model | Fine-tune data | Python | Java | JavaScript | Go | PHP | Ruby | Total/Avg |
> | :--- | :--- | ---: | ---: | ---: | ---: | ---: | ---: | ---: |
> | CodeT5 | CodeSearchNet | 19.55 | 20.38 | 16.15 | 19.83 | 26.26 | 15.38 | **21.24/19.59** |
> |  | TheVault/small | 18.94 | 17.72 | 13.96 | 19.92 | 20.43 | 15.22 | 18.83/17.70 |
> | PLBART | CodeSearchNet | 17.99 | 17.38| 14.84 | 17.98 | 22.54 | 14.08 | **18.78/17.47** |
> |  | TheVault/small | 14.93 | 15.66 | 11.95 | 17.03 | 18.0 | 11.49 | 15.95/14.84 |
>
>
> **Q4. Even though The Vault emphasizes quality, its scale is considerably smaller than some raw datasets like The Stack and The Pile. This may limit its effectiveness in some applications where sheer volume is paramount.**
>
> As you also mentioned, The Vault's goal is to emphasize quality over quantity. Recent work on fine-tuning LLMs also demonstrates that we only need a small amount of high-quality [1] [2] data to fine-tune a model that performs better than that. In our case, The Vault is not intended to replace larger scale datasets such as The Stack and The Pile, but rather to supplement the model that has been trained on such datasets. In our evaluation of code generation, we found that CodeGen [3] when fine-tuned on The Vault outperforms the original pretrained on The Pile and BigQuery (Table 7 in our paper).
>
> **Q5. Given that both CodeSearchNet and The Vault derive significant code samples from GitHub, have you assessed any overlap between the test set of one and the training set of the other to prevent potential data leakage?**
>
> Yes, in our experiment setup, we employed the MinHash LSH technique [4] to filter training instance clusters that are close to samples in the validation and test sets of CodeSearchNet, HumanEval, and MBPP. This filtering was based on code and docstring similarity, in addition to employing exact match deduplication. Additionally, during dataset partitioning into training, testing, and validation sets, we prevented content from the same repository from appearing in multiple sets, thereby avoiding any potential internal data leakage.
>
> **Q6. Recognizing that CodeSearchNet and The Stack are distinct datasets, how might model performance in code search and code summarization tasks differ if we were to draw random text-code pairs from The Stack as a baseline, in comparison to your refined The Vault?**
>
> In response to your concern, we have established 3 function-level subsets, each approximately the size of TheVault/small (~1.7M code-text instances). These subsets were created by randomly sampling the raw function-level dataset extracted from The Stack, without applying any filtering (referred to as raw-TheStack). We use 3 different seeds to sample raw-TheStack and report the average result. The tables below illustrate the impact of our cleaning pipeline, presenting a comparison between models fine-tuned on TheVault/small and those trained on raw-TheStack. These comparisons clearly reveal the filtering methods in The Vault. During training on the raw-TheStack dataset for the code summarization task, we find that the PLBART and CodeT5 generates outputs with substantial noise. These outputs are characterized by a prevalence of special tokens like //; and *. This finding strongly underscores the efficacy of our filtering process in enhancing the quality of the dataset.
>
> *Code Search*
> - Evaluation on TheVault (MRR)
> | Model | Python | Java | JavaScript | Go | PHP | Ruby | Rust | C++ | C | C# |
> | --- | --- | --- | --- | --- | --- | --- | --- | --- | --- | --- |
> | raw-TheStack | 0.318 | 0.3245 | 0.1837 | 0.4194 | 0.1718 | 0.0878 | 0.1949 | 0.2024 | 0.2204 | 0.2309 |
> | TheVault/small | **0.3501** | **0.4214** | **0.3216** | **0.4864** | **0.2351** | **0.2904** | **0.326** | **0.2996** | **0.3015** | **0.3483** |
>
> - Evaluation on CodeSearchNet (MRR)
> | Model | Python | Java | JavaScript | Go | PHP | Ruby |
> | --- | --- | --- | --- | --- | --- | --- |
> | raw-TheStack | 0.3713 | 0.3492 | 0.3148 | 0.5519 | 0.2731 | 0.2748 |
> | TheVault/small | **0.4074** | **0.4857** | **0.4466** | **0.6578** | **0.6578** | **0.5251** |
>
> *Code Summarization:*
> - Evaluate on TheVault (BLEU-4)
> | Model | Fine-tune data | Python | Java | JavaScript | Go | PHP | Ruby | Total/Avg |
> | :--- | :--- | ---: | ---: | ---: | ---: | ---: | ---: | ---: |
> | CodeT5 | raw-TheStack | 16.18 | 9.06 | 6.23| 19.05 | 7.07 | 5.78 | 11.84/10.56 |
> |  | TheVault/small |12.26 | 11.13 | 9.68 | 31.64 | 38.86 | 11.23 | **25.12/19.13** |
> | PLBART | raw-TheStack | 1.69  | 4.02 | 0.43 | 24.60  | 4.83  | 0.49  | 7.19/6.01 |
> |  | TheVault/small | 10.23 | 9.28 | 8.95 | 22.78 | 34.32| 9.74 | **20.29/15.88** |
>
> - Evaluate on CodeSearchNet (BLEU-4)
> | Model | Fine-tune data | Python | Java | JavaScript | Go | PHP | Ruby | Total/Avg |
> | :--- | :--- | ---: | ---: | ---: | ---: | ---: | ---: | ---: |
> | CodeT5 | raw-TheStack | 16.18 | 9.06 | 6.23| 19.05 | 7.07 | 5.78 | 11.84/10.56 |
> |  | TheVault/small | 18.94 | 17.72 | 13.96 | 19.92 | 20.43 | 15.22 | **18.83/17.70**|
> | PLBART | raw-TheStack | 0.86 | 3.06 | 0.59 | 10.91 | 2.29 | 0.47 | 3.23/3.03 |
> |  | TheVault/small | 14.93 | 15.66 | 11.95 | 17.03 | 18.0 | 11.49 | **15.95/14.84** |
>
> **Q7. The term "Multi-TheVault-raw" in Table 7 isn't elaborated upon within the paper. Is this intended to represent a dataset randomly sampled from The Stack?**
>
> Yes, "Multi-TheVault-raw" refers to a specific scenario where the CodeGen-350m-multi model is finetuned on a subset of Python data from The Stack. This subset mirrors the size of Python data present in The Vault dataset. Through this comparison, we demonstrate the advancements achieved in our refinement process and its potential to yield improved results when used for training.
>
> [1] LIMA: Less Is More for Alignment, https://arxiv.org/abs/2305.11206 \
> [2] Instruction Mining: High-Quality Instruction Data Selection for Large Language Models, https://arxiv.org/abs/2307.06290 \
> [3] Nijkamp, E., Pang, B., Hayashi, H., Tu, L., Wang, H., Zhou, Y., ... & Xiong, C. (2022, September). CodeGen: An Open Large Language Model for Code with Multi-Turn Program Synthesis. In The Eleventh International Conference on Learning Representations. \
> [4] Eric Zhu, & Vadim Markovtsev. (2017). ekzhu/datasketch: First stable release (v1.0.0). Zenodo. https://doi.org/10.5281/zenodo.290602 \
> [5] Liu, Y., Ott, M., Goyal, N., Du, J., Joshi, M., Chen, D., ... & Stoyanov, V. (2019). Roberta: A robustly optimized bert pretraining approach. arXiv preprint arXiv:1907.11692. \
> [6] Guo, D., Lu, S., Duan, N., Wang, Y., Zhou, M., & Yin, J. (2022). Unixcoder: Unified cross-modal pre-training for code representation. arXiv preprint arXiv:2203.03850. \
> [7] Ahmad, W. U., Chakraborty, S., Ray, B., & Chang, K. W. (2021). Unified pre-training for program understanding and generation. arXiv preprint arXiv:2103.06333.

---

### Meta-Review · Senior_Area_Chairs · 2023-10-04

**Recommendation:** 3

**Metareview:**

The paper under consideration introduces "The Vault," a novel dataset designed for training large language models to comprehend and generate code. It consists of 41 million high-quality code-text pairs extracted from 10 prevalent programming languages, surpassing existing datasets in both quality and scale. The authors employ innovative data cleaning techniques, combining rule-based methods with deep learning classifiers, and open-source the toolkit used for dataset creation and quality assurance. The paper provides detailed experiments showcasing the dataset's advantages and is well-written and easy to follow.


**Reasons to Accept:**
1. The paper presents a significantly larger and higher-quality dataset compared to existing alternatives, with empirical evidence demonstrating its superiority in downstream tasks.
2. The authors' data cleaning methodology is innovative and effectively filters out inconsistent code-text pairs.
3. The open-sourcing of the toolkit for dataset creation aligns with the principles of open research and encourages community engagement.
4. The paper's clarity and comprehensive experiments contribute to the understanding of the new dataset's capabilities.


**Reasons to Reject:**
1. The dataset's limitation to 10 programming languages may restrict its appeal to researchers interested in less prevalent languages.
2. The evaluation of code generation relies on small models, potentially limiting insights into its performance with larger-scale models.
3. The paper's evaluation of the dataset across three tasks using only one model for each task may not provide a holistic assessment of its adaptability across different architectures.
4. The dataset, while high-quality, is smaller in scale compared to some raw datasets, potentially limiting its effectiveness in applications requiring larger volumes of data.

The paper appears to be sound, with reviewers generally agreeing on its methodological rigor and contributions. However, excitement levels are mixed, with some reviewers expressing reservations about the novelty and impact of the work.

---

### Decision · Program_Chairs · 2023-10-07

**Decision:**

Accept-Findings

**Comment:**

The paper under consideration introduces "The Vault," a novel dataset designed for training large language models to comprehend and generate code. It consists of 41 million high-quality code-text pairs extracted from 10 prevalent programming languages, surpassing existing datasets in both quality and scale. The authors employ innovative data cleaning techniques, combining rule-based methods with deep learning classifiers, and open-source the toolkit used for dataset creation and quality assurance. The paper provides detailed experiments showcasing the dataset's advantages and is well-written and easy to follow.


**Reasons to Accept:**
1. The paper presents a significantly larger and higher-quality dataset compared to existing alternatives, with empirical evidence demonstrating its superiority in downstream tasks.
2. The authors' data cleaning methodology is innovative and effectively filters out inconsistent code-text pairs.
3. The open-sourcing of the toolkit for dataset creation aligns with the principles of open research and encourages community engagement.
4. The paper's clarity and comprehensive experiments contribute to the understanding of the new dataset's capabilities.


**Reasons to Reject:**
1. The dataset's limitation to 10 programming languages may restrict its appeal to researchers interested in less prevalent languages.
2. The evaluation of code generation relies on small models, potentially limiting insights into its performance with larger-scale models.
3. The paper's evaluation of the dataset across three tasks using only one model for each task may not provide a holistic assessment of its adaptability across different architectures.
4. The dataset, while high-quality, is smaller in scale compared to some raw datasets, potentially limiting its effectiveness in applications requiring larger volumes of data.

The paper appears to be sound, with reviewers generally agreeing on its methodological rigor and contributions. However, excitement levels are mixed, with some reviewers expressing reservations about the novelty and impact of the work.